# Influence of Coexistence of Pitting and Cracking Faults on a Two-Stage Spur Gear System

**Kemajou Herbert Yakeu Happi** *, **Bernard Xavier Tchomeni Kouejou** and **Alfayo Anyika Alugongo**

Department of Industrial Engineering, Operation Management and Mechanical Engineering,
Vaal University of Technology, Vanderbijlpark 1900, South Africa
* Correspondence: kemajouy@vut.ac.za

**Abstract:** This work considers forced vibrations in a rotating structure consisting of a two-stage spur gear system with coexisting defects, specifically pitting and cracking. Numerical simulations and experimental analysis in various scenarios of the system in operation were conducted using the RPM–Frequency mapping technique. To identify fault characteristics, the analysis performed assumed the gear system had been misadjusted by a combination of pitting and cracking on the gear teeth. The correlation of the system-forced responses under regular and chaotic vibrations revealed that the system is far more sensitive to the crack than to the pitting when there are fluctuating harmonic peaks present at high vibration levels.

**Keywords:** two-stage gear system; pitting and crack faults; RPM-frequency map

## 1. Introduction

Gear transmission maintenance is crucial for extending the life and reliability of equipment. However, the unique design and demanding working conditions of gear systems make predicting failure and damage difficult. Transmission errors can cause increased noise and vibration, and defects on the gear teeth can weaken resistance and lead to destruction [1]. Pitting and cracking are common gear problems that can cause significant failures. It is important to study these issues to monitor the condition of the transmission system and diagnose faults early.

Top of Form

Cracks in the gear teeth affect the bending stiffness due to stress in the root that exceeds the material endurance limit, but do not affect contact stiffness [2]. Progressive cracks appear at the base of the tooth defect with each rotation of the mechanism, and are particularly visible in thin, heat-hardened stainless steels that are highly stressed.

Pitting can cause the deformation of the teeth and the gear transmission system, resulting in vibration and noise problems.

These crack and pitting defects, when not detected early, significantly influence the vibration behavior of the gear system [3]. Therefore, researching the mechanism and diagnosing gear defects is crucial to avoid problems.

In the literature, several publications on monitoring and diagnosing gear systems have addressed cracks and pitting separately, with few addressing the vibration signature of coexisting nonlinear cracks and pitting in a two-stage spur gear, which is a difficult problem to diagnose and analyze.

Condition monitoring and fault diagnostics are useful for ensuring the safe running of machines [4]. To meet the ever-increasing demand for the maintenance of gear systems, industrial companies have traditionally depended on the shutdown of the machines or processing the fault diagnosis. However, online monitoring has proven to be effective in terms of machine state analysis and fault prediction. Vibration-based condition monitoring

has become increasingly important in the maintenance of industrial and automotive gearboxes [5]. Vibration analysis can be used to detect damage in gear systems by analyzing the vibrations produced by the gears as they rotate.

Analysis of the vibration characteristics of cracked gears provides a theoretical basis for diagnosing cracking faults. The potential energy approach was initially developed for calculating gear stiffness by Yang and Lin [6]. Based on this approach, Wei investigated the calculation of the time-varying meshing stiffness and dynamic characteristics of a two-stage helical gear system [7]. Hertz's contact energy, bending deformation energy, and axial compression energy all contributed to the elastic deformation of the gear. The gear is seen as a variable cross-section cantilever beam in this method. Meng created a dynamic model with a cracked first-level spur gear, using the segmental stiffness approach to examine the vibrational response of the crack rupture [8]. Chen also created a dynamic model to study the dynamic properties of a two-stage planetary gear transmission with a fractured gear [9].

Yan et al. [10] used an analytical gear meshing stiffness calculation model with tooth surface crack. Li et al. [11] considered the axial force of a helical gear as an improved calculation method for TVMS with crack faults.

Methods for diagnosing and identifying gear pitting defects have been divided into two categories in recent years: model-based and data-based [12].

Ma et al. applied a similar model, but their study focused on the impact of pitting evolution on dynamic gear responses [13]. Meng et al. created a model to examine the changes in gear meshing stiffness as severe pitting levels grew from healthy gearing, modeling the geometry of the pits as a sphere [14]. Pitting is caused by friction between the tooth surfaces during the gear meshing process; however, the cracking expands with each setting load and is located at the tooth root. The effective contact area is reduced, the gear teeth are bent, and the gear transmission system's bearing capacity is reduced because of pitting on the surface of the gear teeth only. Its appearance is due to exceeding the elastic limit in stress at the root of the tooth and on the side of the tooth in tension. Because of the change in the meshing stiffness of the gear, the deformation of the teeth and gear transmission system is more likely to cause vibration in the working process, resulting in system vibration and noise problems.

Hou et al. [15] used the potential energy approach to solve the TVMS of ideal gears with various levels of pitting severity.

For the diagnosis and analysis of the two defects, time–frequency analysis is a modern diagnostic approach with great sensitivity that gives a good diagnostic capacity for characterizing the dynamic behavior of the gear system [16–18]. The set of coexistence defects present on the gear teeth weakens the mechanical resistance, which can lead to their destruction and cause enormous damage. Model-based techniques require a thorough understanding of dynamic modeling as well as precise system condition parameters, and vibration analysis is the most extensively utilized technique [19].

Indeed, by utilizing the appropriate technology to provide qualitative information on the forces applied to the machine components, this method enables the diagnosis of a possible malfunction. Spectral analysis is the approach used most frequently to investigate mechanical vibration. Furthermore, depending on the critical method of determining the intensity of the damage in local fault detection, the symptoms of the fault can be determined in the experimental instance. However, the application of such a technique in the analysis of multiple combined nonlinear faults is still a subject of study where predicting the maintenance of the gear system and preventing an untimely stoppage in a machine process are concerned. In recent years, based on the method mentioned above, the TVMS of gears without fault has been obtained by many researchers [20], and many scholars have studied the tooth crack fault of gears [8,21]. Tiancheng et al. [22] used a three-dimensional finite element model to derive and validate the gear mathematical model with pitting–crack coupling faults.

The pitting and cracking of gear teeth can be understood by looking at the mechanics of the gear meshing process. When gears mesh, the teeth meet each other at a specific

point, called the contact point. At this point, the gears experience high contact stresses and pressure, as the teeth must conform to each other in order to transmit power.

Pitting can occur when these high contact stresses and pressure cause small surface defects, called pits, to form on the gear teeth. These pits can be caused by a variety of factors, including improper lubrication, high loads, and poor surface finish.

Cracking can occur when the high contact stresses and pressure cause small surface cracks to form on the gear teeth, which can then become larger over time. These cracks can be caused by a variety of factors, including improper heat treatment, high loads, and poor surface finish.

Both pitting and cracking can reduce the strength and durability of the gears and can ultimately lead to gear failure if left unaddressed [23].

This paper proposes a novel model of tooth-pitting–crack coexistence faults. A method for calculating the stiffness of a spur gear with combined pitting and crack faults is also proposed. The influence of crack and pitting parameters and time-varying mesh stiffness on the vibration characteristics of a gear system is studied. The coexistence of pitting and cracking faults on a two-stage spur gear system is analyzed. The time domain and frequency domain results and the RPM–frequency map are obtained for the 10 degrees of freedom gear system with coexisting defects and compared with experimental results. The results obtained in the first stage serve as the baseline for the second stage of the study, which examines the behavior of the vertical vibrations generated by the gear system.

This paper is divided into the following sections: Section 2 introduces the proposed calculation model of gear meshing stiffness for a gear with coupled pitting and cracking on the tooth surface. Section 3 provides a brief overview and the governing equation for the dynamic mathematical modeling of a two-stage spur gear system, while in Section 4 the description of the test bench and experimental findings for validating the simulated response using the proposed dynamic model are shown. Section 5 discusses the two-stage gear system fault characteristics in relation to simulation and experiment. The significant findings and conclusions of the work are highlighted in Section 6.

## 2. Calculating Gear Meshing Stiffness Model

In this study, based on the parameters of a spur gear system shown in Table 1, the calculation of the meshing stiffness of the spur gear begins by illustrating the perfect state of the gear, followed by the damaged one.

**Table 1.** The main features of spur gears.

| Parameters | Value | |
| --- | --- | --- |
| | Driving Gear (Pinion) [1,3] | Driven Gear (Wheel) [2,4] |
| Young Module (E) [Pa] | $2.068 \times 10^{11}$ | $2.068 \times 10^{11}$ |
| Pressure angle (°) | 20 | 20 |
| Poisson's ratio | 0.3 | 0.3 |
| Number of teeth $Z_1 = Z_3$ (pinion) and $Z_2 = Z_4$ (gear) | 30 | 90 |
| Base circle radius of a pinion $R_1 = R_3$ [mm] and gear $R_2 = R_4$ [mm] | 30.1 | 76.1 |
| Mass $m_1$ (pinion) and $m_2$ (gear) [kg] | 0.96 | 2.88 |
| Meshing stiffness of bearings $k_1$, $k_3$ (pinion) = $k_2$, $k_4$ (gear) [N.s/m] | $6.56 \times 10^7$ | $6.56 \times 10^7$ |
| Damping coefficient of bearings $c_1$, $c_3$ (pinion) = $c_2$, $c_4$ (gear) [N.s/m] | $1.8 \times 10^5$ | $1.8 \times 10^5$ |
| Torsional stiffness of coupling $k_p$ (pinion) = $k_g$ (gear) [N.s/m] | $4.4 \times 10^4$ | $4.4 \times 10^4$ |
| Damping coefficient of coupling $c_p$ (pinion) = $c_g$ (gear) [Nm. s/rad] | $5 \times 10^5$ | $5 \times 10^5$ |

### 2.1. Spur Gear at the Perfect State

This section focuses on the correct gears, the involute of the circles, and the numerical model vibration responses. An involute of a circle is defined as the curve created by a point N on a straight line as it rolls without sliding on a circle, known as base circle (Figure 1). The gear system is evaluated with a meshing stiffness equal to that of a pair of involute gear profiles with no machining errors.

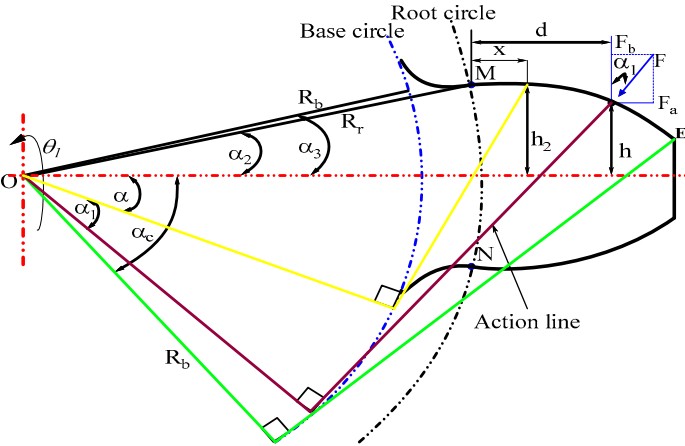

**Figure 1.** Beam model for a spur gear tooth when the root circle is bigger than the base circle.

This section determines the effective gear meshing stiffness using the potential energy principle.

The total potential energy stored in a gear system is composed of four components, including the Hertzian energy, bending energy, axial compression energy, and shear energy, which are expressed as follows:

$$U_h = \frac{4(1 - v^2)}{\pi EL} \tag{1}$$

$$U_b = \frac{1}{2}\frac{F^2}{k_b} = \int_0^d \frac{(F_b(d - x) - M)^2}{2EI_x}dx \qquad 0 < x < d \tag{2}$$

$$U_a = \frac{1}{2}\frac{F^2}{k_a} = \int_0^d \frac{F_a^2}{2EA_x}dx \qquad 0 < x < d \tag{3}$$

$$U_s = \frac{1}{2}\frac{F^2}{k_s} = \int_0^d \frac{1.2F_b^2}{2GA_x}dx \qquad 0 < x < d \tag{4}$$

where the moment of inertia of the surface $I_x$ and section area $A_x$ of the tooth can be calculated as follows:

$$I_x = \frac{1}{12}(2h_x)^3 L \tag{5}$$

$$A_x = 2 \times h_x \times L \tag{6}$$

and the expression of which can be determined using the following formula:

$$G = \frac{E}{2(1 + v)} \tag{7}$$

As illustrated in Figure 1, the bending force $F_b$ produces bending and shear effects, while the axial force $F_a$ produces axial compression and bending effects. Couple M was used to illustrate the bending effect $F_a$. It is expressed as follows:

$$M = F_a \times h \tag{8}$$

where h is the distance between the point of contact and the tooth center line, which may be calculated using the following formula [8]:

$$h = R_b[(\alpha_1 + \alpha_2)\cos\alpha_1 - \sin\alpha_1] \tag{9}$$

$\alpha_2$ is half the angle of the tooth base.

To study the properties of the parameters for the angular displacements of the pinion or wheel (angular displacement is the angle made by the pinion or wheel with respect to a reference point), it is more convenient to express mathematical relationships in terms of an angular rather than a linear variable (reflecting displacement). The involute geometry of the gear profile is expressed using the following formula:

$$x = R_b\cos\alpha - R_b(\alpha_2 - \alpha)\sin\alpha - R_b\cos\alpha_2 - R_r\cos\alpha_3 \tag{10}$$

whereas the height of the section, the distance between the involute corresponding to the section at a distance from the root of the tooth, and the center line of the tooth can be calculated as follows:

$$h_x = R_b[(\alpha_2 - \alpha)\cos\alpha + \sin\alpha] \tag{11}$$

where $\alpha$ is the gear rotation angle.

Depending on the geometry of the involute tooth in Figure 1, the distance d between the contact point and the tooth root can be expressed as follows:

$$d = R_b[(\alpha_1 + \alpha_2)\sin\alpha_1 + \cos\alpha_1] - R_r\cos\alpha_3 \tag{12}$$

The expression of the term $(d - x)$ can be obtained by subtracting the Equations (10) and (12):

$$d - x = R_b(\alpha_1\sin\alpha_1 + \alpha_2\sin\alpha_1 + \cos\alpha_1 - \cos\alpha + \alpha_2\sin\alpha + \alpha\sin\alpha) \tag{13}$$

where $\alpha_2$ is the half-tooth angle on the base circle, $\alpha_3$ describes the approximated half-tooth angle on the root circle, and $\alpha$ is the angular displacement, expressed as follows:

$$\alpha_2 = \frac{\pi}{2Z_1} + \tan\alpha_0 + \alpha_0\text{the} \tag{14}$$

$$\alpha_3 = \arcsin\left(\frac{R_b\sin\alpha_2}{R_r}\right) \tag{15}$$

where $Z_1$ is the number of teeth of the pinion. The expression of $\alpha_1$ is expressed as follows:

$$\alpha_1 = \tan\left(\arccos\frac{Z_2\cos\alpha_0}{Z_2 + 2}\right) - \frac{\pi}{2Z} - \tan\alpha_0 + \alpha_0 - \frac{Z_1}{Z_2}\alpha \tag{16}$$

where $Z_1$ and $Z_2$ are the numbers of teeth of the gear and pinion and $\alpha_0$ is the pressure angle.

The effective gear meshing stiffness is only considered when analyzing the coupled torsional–lateral vibrations of pair spur gears in a one-stage gearing system to identify pitting and cracking and simplify stiffness calculation.

## 2.2. Cracking and Pitting into the Tooth Surface

In this study, a cantilever beam model is integrated with single pitting and general cracking calculations to simulate the coupling of two faults. This is illustrated in Figure 2c.

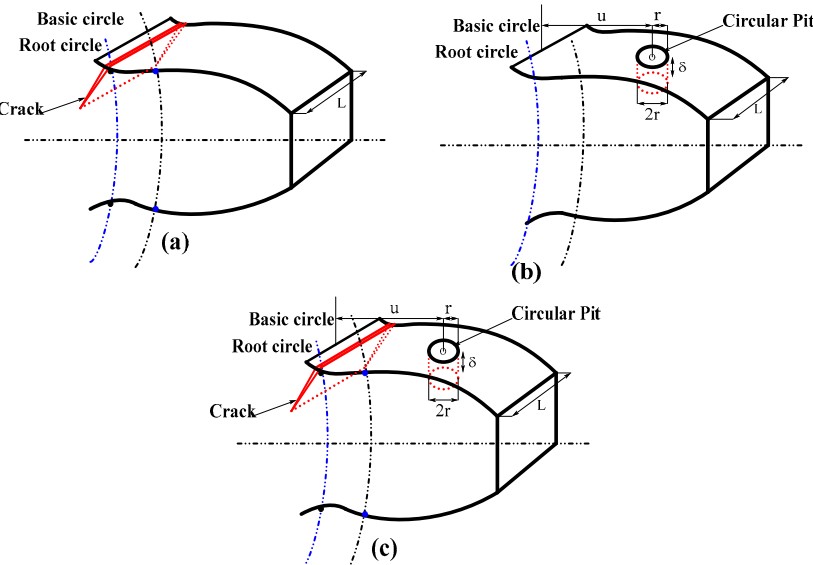

**Figure 2.** Isometric profile of the tooth showing: (**a**) crack; (**b**) circular pit; and (**c**) coexistence of pitting and cracking.

The study of the coexistence of pitting and cracking in gears starts by showing the path of crack propagation at the root of the tooth, defined as a straight line in Figure 2a, and then the normal distribution of pitting along the tooth profile direction and the uniform distribution of pitting along the tooth width (L) direction, as shown in Figure 2b.

Three variables are used to express the circular pitting: the pitting depth ($\delta$), the distance between the tooth root and the pitting circle center (u), and the circle radii of the pitting (r). Furthermore, the different top views of the affected gear tooth are presented as shown in Figure 3. At the moderate stage, Figure 3a shows the tooth failure region with a constant crack depth, and Figure 3b shows the tooth failure region with a pit depth. As a result, the coexistence of pitting and cracking is uniformly distributed in Figure 3c. This changes the effective section of the gear tooth area and the area moment of inertia, as well as the gear meshing stiffness.

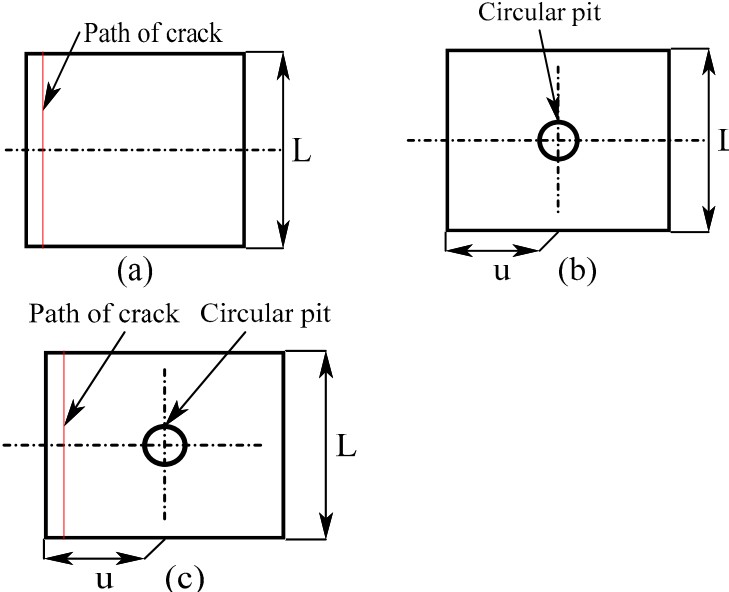

**Figure 3.** Top view of the tooth showing: (**a**) crack; (**b**) circular pit; and (**c**) coexistence of pitting and cracking.

The most widely used model in studies of gear meshing stiffness is the cantilever beam model. In this study, calculations for single pitting and cracking are combined with the cantilever beam model, as shown in Figure 4a, to simulate the coexistence of the two faults. Figure 4b shows the affected tooth area at a moderate stage, with the maximum stress on a spur gear tooth occurring at the point of contact between the two gear teeth. Therefore, the location of the crack and pit, as shown in Figures 2 and 3, can have a significant impact on the stress on the tooth.

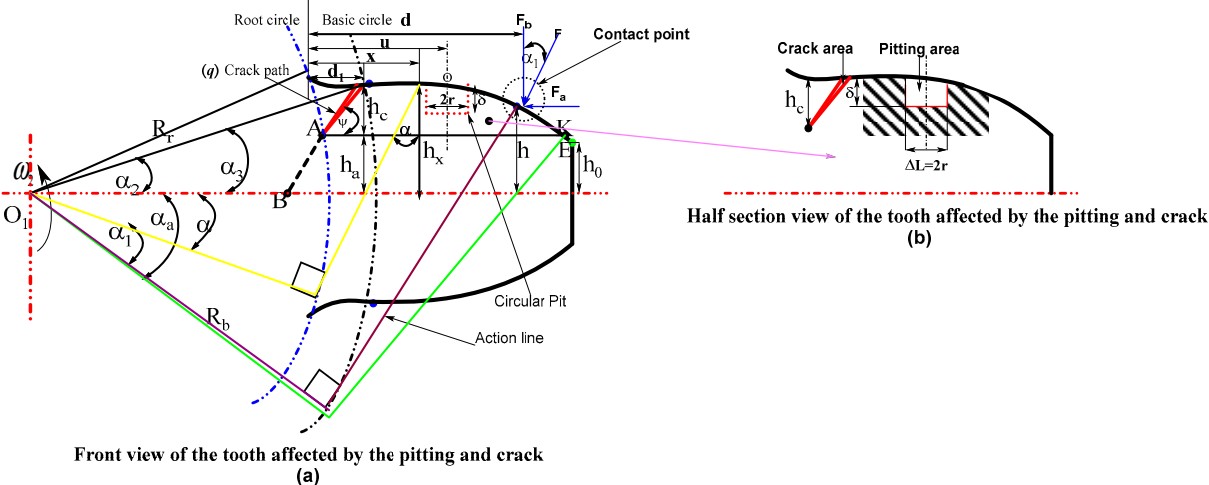

**Figure 4.** Tooth cantilever beam model of faulty gear: (**a**) front view and (**b**) haft view.

Accordingly, the sectional area through pitting and cracking will change both the area and the area moment of inertia of the effective section, leading to different effects of the combined crack and pitting on the meshing stiffness.

The system vibrates due to the contact of the loads at the meshing, which fluctuates in response to the point of contact movement along the line of action. The transition is primarily caused by a single tooth pair and variations in mesh stiffness during contact, either without or with the pitting or crack fault. As a result, it is essential to investigate this fluctuation in stiffness during the rotation of gears.

Due to the crack's influence on the effective moment of inertia and the cross-section of the surface, the bending and shear stiffness will change. As a result, the effective moment of inertia and cross-section of the surface at a distance x (Equation (10)) from the root of the tooth are determined using Equations (17) and (18).

$$I_{xa} = \begin{cases} \frac{1}{12}(h_a+h_x)^3 L & if \ \ x \le d \\ \frac{1}{12}(2h_x)^3 L & if \ \ x > d \end{cases} \tag{17}$$

$$A_{xa} = \begin{cases} (h_a+h_x) L & if \ \ x \le d \\ (2h_x) L & if \ \ x > d \end{cases} \tag{18}$$

where $I_{xa}$ is the effective moment of inertia of the cracked tooth, $A_{xa}$ is the cross-section of the surface of the cracked tooth, and **$h_a$** is the height of the section at point A of the crack, which can be written as follows:

$$h_a = R_b \sin\alpha_2 - q\sin\psi \tag{19}$$

The crack depth called q is only considered if it is less than half the thickness of the tooth base noted h0, and $\psi$ the crack angle. Hx is denoted as the height of the section at the lowest contact point E on the tooth.

### 2.2.1. Derivation of Mesh Stiffness for Cracked Gears

From the equations above, 2, 3, and 4 become the integration variable rather than x. The bending stiffness equation can be obtained by using the potential energy of the deflection Equation (1), which is based on the beam theory,

Case 1: when $h_a < h_0$ or $h_a \geq h_0$ and $\alpha_1 < \alpha_a$

$$U_{b_{crack}} = \frac{1}{2}\frac{F^2}{k_{b_{crack}}} = \int_0^d \frac{(F_b(d-x)-M)^2}{2EI_{xa}}dx \tag{20}$$

$$\frac{1}{k_{b_{crack}}} = \int_0^d \frac{[\cos\alpha_1(d-x)-\sin\alpha_1 h]^2}{EI_{xa}}dx \tag{21}$$

By deriving the expression of x in terms of $\alpha$, its derivative is expressed as follows:

$$\frac{dx}{d\alpha} = -R_b\sin\alpha - (R_b(-1)\sin\alpha - R_b(\alpha_2-\alpha)\cos\alpha) \tag{22}$$

$$dx = R_b(\alpha-\alpha_2)\cos\alpha d\alpha \tag{23}$$

Substituting Equation (20) into the term Equation (17), $\alpha$ becomes the integration variable rather than x:

$$\frac{1}{k_{b_{crack}}} = \int_{-\alpha_c}^{\alpha_2} \frac{R_b^3[1+\cos\alpha_1(-\cos\alpha+(\alpha_2+\alpha)\sin\alpha]^2(\alpha-\alpha_2)\cos\alpha}{E.I_{xa}}d\alpha \tag{24}$$

where is the expression of the effective moment of inertia substituted into the Equation (21).

Therefore, the bending stiffness equation is expressed as follows:

$$\frac{1}{k_{b_{crack}}} = \int_{-\alpha_c}^{\alpha_2} \frac{12[1+\cos\alpha_1(-\cos\alpha+(\alpha_2+\alpha)\sin\alpha)]^2(\alpha-\alpha_2)\cos\alpha}{E.L\left(\sin\alpha_2 - \frac{q}{R_b}\sin\psi + (\alpha_2-\alpha)\cos\alpha + \sin\alpha\right)^3}d\alpha \tag{25}$$

Case 2: When $h_a \geq h_o$ and $\alpha_1 > \alpha_a$
$I_x$ remains the same as in the perfect state.

$$I_x = \frac{2R_b^3}{3}[((\alpha_2-\alpha)\cos\alpha + \sin\alpha)]^3 L \tag{26}$$

When the crack is above the tooth center line $(-\alpha_a < \alpha < -\alpha_2)$
$I_{xa}$ remains the same as in the first case.

$$I_{xa} = \frac{R_b^3}{12}\left[\sin\alpha_2 - \frac{q}{R}\sin\psi + ((\alpha_2-\alpha)\cos\alpha + \sin\alpha)\right]^3 L \tag{27}$$

The potential energy of the deflection (bending) can be calculated using beam theory, taking into account both set limitations with and without crack:

$$\frac{1}{k_{b_{crack}}} = \int_{-\alpha_c}^{\alpha_2} \frac{3[1+\cos\alpha_1(-\cos\alpha+(\alpha_2+\alpha)\sin\alpha)]^2(\alpha_2+\alpha)\cos\alpha}{2.E.L((\alpha_2-\alpha)\cos\alpha+\sin\alpha)^3}d\alpha +$$
$$\int_{-\alpha_a}^{\alpha_2} \frac{12[1+\cos\alpha_1(-\cos\alpha+(\alpha_2+\alpha)\sin\alpha)]^2(\alpha_2+\alpha)\cos\alpha}{E.L\left(\sin\alpha_2-\frac{q}{R_b}\sin\psi+(\alpha_2-\alpha)\cos\alpha+\sin\alpha\right)^3}d\alpha \tag{28}$$

When the crack is below the tooth center line $(-\alpha_c < \alpha < -\alpha_a)$
$A_x$ remains the same as in the perfect state.

$$A_x = 2(R_b[(\alpha_2-\alpha)\cos\alpha + \sin\alpha])L \tag{29}$$

When the crack is above the tooth center line $(-\alpha_a < \alpha < \alpha_2)$

$A_{xa}$ remains the same as in the first case.

$$A_{xa} = R_b\left(\sin\alpha_2 - \frac{q}{R_b}\sin\psi + (\alpha_2 - \alpha)\cos\alpha + \sin\alpha\right)L \tag{30}$$

The shear potential energy is used to obtain the expression of the shear stiffness equation, which is based on beam theory:

$$\frac{1}{k_{s_{crack}}} = \int_{-\alpha_c}^{-\alpha_a} \frac{1.2(1+v)\cos^2\alpha_1(\alpha_2+\alpha)\cos\alpha}{E.L((\alpha_2-\alpha)\cos\alpha+\sin\alpha)}d\alpha + \int_{-\alpha_a}^{\alpha_2} \frac{1.2(1+v)\cos^2\alpha_1(\alpha_2+\alpha)\cos\alpha}{E.L\left(\sin\alpha_2 - \frac{q}{R_b}\sin\psi + (\alpha_2-\alpha)\cos\alpha+\sin\alpha\right)}d\alpha \tag{31}$$

Then, because of the crack effect, the bending and shear stiffness equations change. As a result of the presence of the crack in the tooth root, the total meshing stiffness of a pair of spur gears is composed of the Hertzian, bending, shear, and axial compressive stiffness in sequence and can be expressed as follows:

$$k_{t(crack)} = \frac{1}{\frac{1}{k_h} + \frac{1}{k_{a_1}} + \frac{1}{k_{b_1(crack)}} + \frac{1}{k_{s_1(crack)}} + \frac{1}{k_{a_2}} + \frac{1}{k_{b_2}} + \frac{1}{k_{s_2}}} \tag{32}$$

2.2.2. Derivation of Mesh Stiffness for Pitted Gears

The potential energy approach is used to calculate the effect of a pitted tooth on the gear meshing stiffness. As shown in Figure 2b, a pinion tooth is described as a non-uniform cantilever beam starting from the root circle.

The effective tooth contact width L is not constant and decreases when the surface of the pinion tooth has pitting dispersed over adjacent teeth during meshing. Its expression changes $(L - \Delta L_x)$ correspondingly, and affects the expressions of $h_x$, $I_x$, and $A_x$, all different from the ones given above for a perfect pinion tooth. $\Delta L_x$, $\Delta A_x$, and $\Delta I_x$ are used to represent the reduction of the tooth's contact width, area, and area moment of inertia, where x is the distance to the tooth root. Their expressions are calculated as follows:

$$\Delta L_x = \begin{cases} 2\sqrt{r^2 - (u-x)^2}, & x \in [u-r, u+r] \\ 0, & \text{others} \end{cases} \tag{33}$$

$$\Delta A_x = \begin{cases} \Delta L_x \delta, & x \in [u-r, u+r] \\ 0, & \text{others} \end{cases} \tag{34}$$

$$\Delta I_x = \begin{cases} \frac{1}{12}\Delta L_x \delta^3 + \frac{\left(A_x \Delta A_x \left(h_x - \frac{\delta}{2}\right)\right)^2}{A_x - \Delta A_x} & x \in [u-r, u+r] \\ 0 & \text{others} \end{cases} \tag{35}$$

where u represents the distance between the tooth root and the circle centre of the pit, r is the radius of the pitting circle, and δ is the pitting depth.

Then, for gear pairs with a circular tooth pit, the Hertzian contact stiffness $k_h$, bending stiffness $k_b$, axial stiffness $k_a$, shear stiffness $K_s$, and Hertzian contact stiffness are deduced as follows:

$$k_h = \frac{\pi E(L - \Delta L_x)}{4(1-\mu)} \tag{36}$$

$$\frac{1}{k_{b(pitting)}} = \frac{\left[1 - \frac{(Z-2.5)(\cos\alpha_1\cos\alpha_2)}{Z\cos\alpha_0}\right]^3 - (1-\cos\alpha_1\cos\alpha)^3}{2EL\cos\alpha_1 + \sin^3\alpha} + \int_{-\alpha_1}^{\alpha_2} \frac{3[1 + \cos\alpha_1(-\cos\alpha + (\alpha_2-\alpha)\sin\alpha)]^2(\alpha_2-\alpha)\cos\alpha}{E\left[2L((\alpha_2-\alpha)\cos\alpha+\sin\alpha)^3 - 3\frac{\Delta L_x}{R_b}\right]}d\alpha \tag{37}$$

$$\frac{1}{k_{s(Pitting)}} = \frac{1.2(1+v)\cos^2\alpha_1\left(\cos\alpha_2 - \frac{Z-2.5}{Z\cos\alpha_0}\cos\alpha_3\right)}{EL\sin\alpha_2} + \int_{-\alpha_1}^{\alpha_2} \frac{1.2(1+v)\cos^2\alpha_1(\alpha_2+\alpha)\cos\alpha\cos^2\alpha_1}{E\left[L((\alpha_2-\alpha)\cos\alpha+\sin\alpha) - \frac{1}{2}\frac{\Delta A_x}{R_b}\right]}d\alpha \tag{38}$$

$$\frac{1}{k_{a_{(Pitting)}}} = \frac{\sin^2\alpha_1\left(\cos\alpha_2 - \frac{Z-2.5}{Z\cos\alpha_0}\cos\alpha_3\right)}{2ELsin\alpha_2} + \int_{-\alpha_1}^{\alpha_2} \frac{(\alpha_2+\alpha)\cos\alpha\sin^2\alpha_1}{E\left[2L((\alpha_2-\alpha)\cos\alpha+\sin\alpha)-\frac{\Delta A_x}{R_b}\right]}\,d\alpha \quad (39)$$

The total effective meshing stiffness of a pair of spur gears is expressed by the series combination of bending stiffness $k_b$, axial stiffness $k_a$, shear stiffness $k_s$, and Hertzian contact stiffness $k_h$ as follows:

$$k_{t_{pitted}} = \frac{1}{\frac{1}{k_{h(pit)}} + \frac{1}{K_{b_1(pit)}} + \frac{1}{K_{s_1(pit)}} + \frac{1}{k_{a_1(pit)}} + \frac{1}{k_{b_2}} + \frac{1}{k_{s_2}} + \frac{1}{k_{a_2}}} \quad (40)$$

### 2.2.3. Derivation of Total Effective Gear Mesh Stiffness under Coexistence of Pitting and Cracking

Considering the first stage of the spur gear system affected by the coexistence of the crack and pits in the pinion tooth root and surface, the Hertzian, bending, shear, and axial compression stiffnesses change accordingly to $k_{h(pit)}$, $k_{b_1(pit+crack)}$, $k_{s_1(pit+crack)}$, and $k_{a_1(pit)}$ in sequence. As a result, the expression for total effective mesh stiffness for a two-stage spur gear is:

$$k_{total} = \left[\underbrace{\underbrace{\frac{1}{\frac{1}{k_{h(pit)}} + \frac{1}{k_{b_1(pit+crack)}} + \frac{1}{k_{s_1(pit+crack)}} + \frac{1}{k_{a_1(pit)}}}_{\text{damaged driving gear(pinion)}} + \underbrace{\frac{1}{k_{b_2}} + \frac{1}{k_{s_2}} + \frac{1}{k_{a_2}}}_{\text{healthy driven gear}}}_{k_{t1}\,@\text{first stage}}\right] + \left[\underbrace{\frac{1}{\frac{1}{k_h} + \underbrace{\frac{1}{k_{a_3}} + \frac{1}{k_{b_3}} + \frac{1}{k_{s_3}}}_{\text{health driving gear(pinion)}} + \underbrace{\frac{1}{k_{a_4}} + \frac{1}{k_{b_4}} + \frac{1}{k_{s_4}}}_{\text{healthy driven gear}}}}_{k_{t2}\,@\text{second stage}}\right] \quad (41)$$

### 2.3. Gear Mesh Stiffness Evaluation with Coupled Pitting and Cracking on the Tooth Surface

The equations mentioned are used to compare the mesh stiffness of a pair of spur gears under 25% of pitting (moderate), 29% of cracking (moderate), and the coexistence of pitting and cracking as displayed in Figure 5.

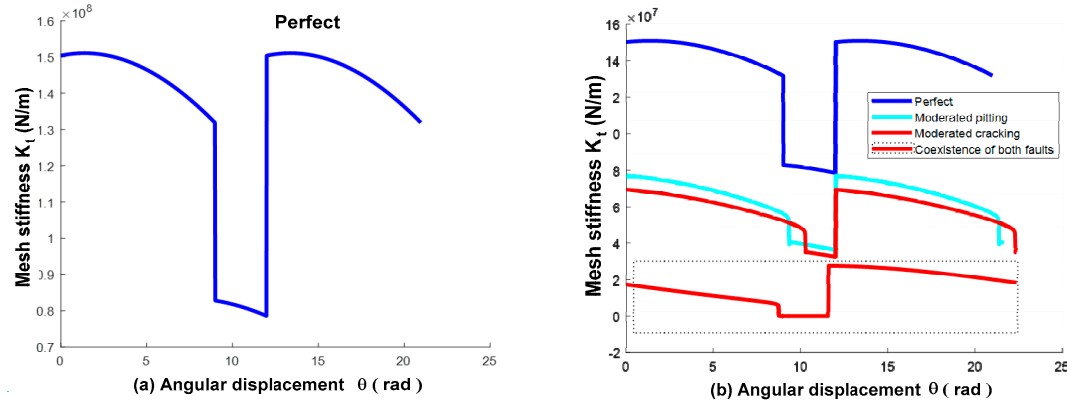

**Figure 5.** Mesh stiffness: (**a**) healthy and (**b**) pitting–crack gear.

The results in Figure 5a demonstrate no irregularities in the pinion and wheel profiles, as well as no reduction in the gear meshing stiffness. In Figure 5b, for a moderate pitting of 25% on the tooth surface, the meshing stiffness varies from $15 \times 10^7$ to $7.772 \times 10^7$ N/m at a period of 21.45 s. However, for a moderate cracking of 29% on the tooth root, the meshing

stiffness decreases from $7.772 \times 10^7$ to $6.9 \times 10^7$ N/m at a period of 22.31 s, which implies significant damage to the major components of the gear vibration signal.

Due to the coexistence of pitting and cracking, the mesh stiffness for the severe fault varies from $6.9 \times 10^7$ to $1.732 \times 10^7$ N/m. Pitting and cracking together enhance gear bending fatigue and increase the weakening of the fractured tooth by dramatically lowering gear mesh stiffness, and it was noted that the crack fault is more dominant on the gear mesh stiffness than the pitting.

## 3. Spur Gear Dynamic Response

Figure 6a shows a two-stage gearbox with six bearings used to support the three shafts—shaft 1 (input), rotating at ($T_m$) revolution per minute, which carried a pinion of mass 1, a base circle radius 1, and moment of inertia 1 ($m_1$, $R_1$, $J_1$) driven by an electric motor via an input coupling joint, which has the following characteristics: a torsional stiffness ($k_p$) and a damping coefficient ($c_p$); shaft 2 (intermediary), which carried a wheel of mass 2, base circle radii 2, within a moment of inertia 2 ($m_2$, $R_2$, $J_2$), and carried a pinion of mass 3, base circle radius 3, and moment inertia 3 ($m_3$, $R_3$, $J_3$); and shaft 2 (output), rotating at ($T_L$) revolution per minute, which carried a wheel of mass 4, a base circle radius 4, and moment of inertia 4 ($m_4$, $R_4$, $J_4$), and was connected to a torsional stiffness ($k_g$) and a damping coefficient ($c_g$) characterizing the load through an output coupling joint, respectively. All bearings are represented by $k_1$, vertical stiffness, and $c_1$, vertical damping on the input bearing; $k_2 = k_3$, vertical stiffness, and $c_2 = c_3$, vertical damping on the intermediary bearing; and k4, vertical stiffness, and c4, vertical damping on the output bearing. Figure 6b,c shows two pairs of spur gears, each with a pinion and a gear. The first stage (b) and second stage (c) are properly coupled, and the two-surface gear contact is subjected to torsional stiffness and damping forces generated by the gear meshing stiffness $k_{t1}$ and damping coefficient $c_{t1}$ in Figure 6b and the gear meshing stiffness $k_{t2}$ and damping coefficient $c_{t2}$ in Figure 6c.

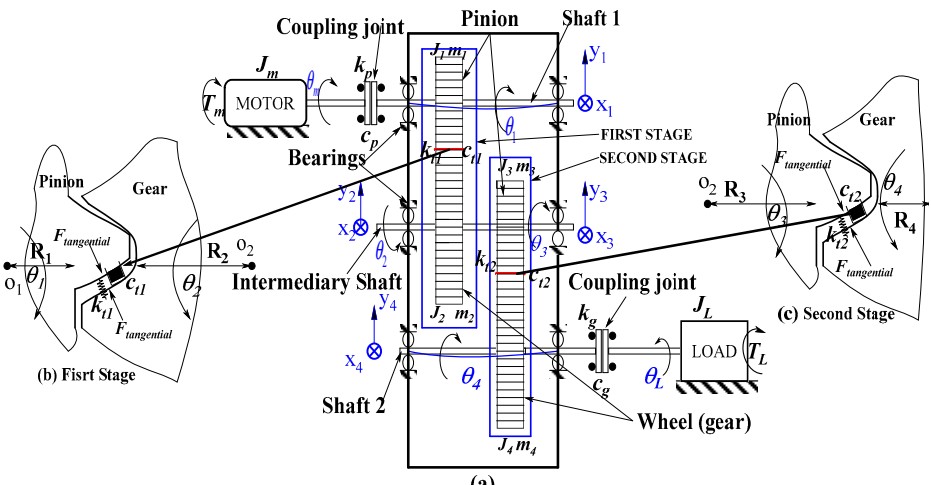

**Figure 6.** Schematic diagram of the two-stage gearbox with 10 DOF. (**a**) Gear transmission system, (**b**) First stage contact, (**c**) Second stage (Pinion-wheel).

Lagrange dynamics in the inertial coordinate system allows us to develop a model system. More detailed information regarding the consideration of adjustment faults such as pitting and cracking, both combined into one fault in the two-stage spur gear system, is given in the following section.

Figure 6a–c, respectively, show the meshing of a pair of spur gears and the diagram of a two-stage spur gear transmission model. The effective gear meshing stiffness is only addressed when analyzing the coupled torsional–lateral vibrations of pair gearing spurs in a two-stage gearing system to identify pitting and cracking to simplify the stiffness

calculation. Cracking and pitting are more likely to develop concurrently and concentrate on the same tooth under heavy load and poor lubrication circumstances, but the mechanism of gear multiple faults is still limited, which presents more obstacles in this study. The model is based on the following premises: (1) the gear body is treated as solid; (2) the shaft is rigid to avoid deformations; (3) engaged teeth are two isotropic elastic bodies; (4) the gear is a standard spur with a pressure angle of 20 degrees; and (6) the root circle is larger than the base circle, or the radius of the circle is greater than the radius of the base (Rr > Rb).

### 3.1. Motion of Dynamic Model

The model proposed in this paper provides a theoretical basis for studying the coexistence of pitting and cracking faults in a two-stage spur gear system and can be used to analyse the vibration characteristics and diagnose the faults.

During the meshing of the teeth, 1 indicates the first pair and 2 indicates the second pair. Based on the geometry of Figure 6, the system's total kinetic energy $T$ is written as follows:

$$T = \tfrac{1}{2}m_1\dot{x}_1^2 + \tfrac{1}{2}m_2\dot{x}_2^2 + \tfrac{1}{2}m_3\dot{x}_3^2 + \tfrac{1}{2}m_4\dot{x}_4^2 + \tfrac{1}{2}m_1\dot{y}_1^2 + \tfrac{1}{2}m_2\dot{y}_2^2 + \tfrac{1}{2}m_3\dot{y}_3^2 + \tfrac{1}{2}m_4\dot{y}_4^2 + \tfrac{1}{2}J_1\dot{\theta}_1^2 + \tfrac{1}{2}J_2\dot{\theta}_2^2 + \tfrac{1}{2}J_3\dot{\theta}_3^2 + \tfrac{1}{2}J_4\dot{\theta}_4^2 + \tfrac{1}{2}J_m\dot{\theta}_m^2 + \tfrac{1}{2}J_L\dot{\theta}_L^2 \tag{42}$$

In the gearing system, the total potential energy $U$, which includes the strain energy of rotating gears, is expressed as follows:

$$U = \tfrac{1}{2}k_1(x_1^2 + y_1^2) + \tfrac{1}{2}k_2(x_2^2 + y_2^2) + \tfrac{1}{2}k_3(x_3^2 + y_3^2) + \tfrac{1}{2}k_4(x_4^2 + y_4^2) + \tfrac{1}{2}k_p(\theta_m - \theta_1)^2 + \tfrac{1}{2}k_g(\theta_4 - \theta_L)^2 + \tfrac{1}{2}k_{t_1}[(y_1 + R_1\theta_1) - (y_2 + R_2\theta_2)]^2 + \tfrac{1}{2}k_{t_2}[(y_3 + R_3\theta_3) - (y_4 + R_4\theta_4)]^2 \tag{43}$$

Because viscous damping is considered in the gearing system, Rayleigh's dissipation function D is expressed as follows:

$$D = \tfrac{1}{2}c_1\left(\dot{x}_1^2 + \dot{y}_1^2\right) + \tfrac{1}{2}c_2\left(\dot{x}_2^2 + \dot{y}_2^2\right) + \tfrac{1}{2}c_3\left(\dot{x}_3^2 + \dot{y}_3^2\right) + \tfrac{1}{2}c_4\left(\dot{x}_4^2 + \dot{y}_4^2\right) + \tfrac{1}{2}c_p\left(\dot{\theta}_m - \dot{\theta}_1\right)^2 + \tfrac{1}{2}c_g\left(\dot{\theta}_4 - \dot{\theta}_L\right)^2 + \tfrac{1}{2}c_{t_1}\left[\left(\dot{y}_1 + R_1\dot{\theta}_1\right) - \left(\dot{y}_2 + R_2\dot{\theta}_2\right)\right]^2 + \tfrac{1}{2}c_{t_2}\left[\left(\dot{y}_3 + R_3\dot{\theta}_3\right) - \left(\dot{y}_4 + R_4\dot{\theta}_4\right)\right]^2 \tag{44}$$

where the driving pinions are 1 and 3 and the driven gears are 2 and 4, respectively.

In this model of the two-stage spur gear system, 10 DOF are considered with four lateral displacements $y_1$, $y_2$, $y_3$, and $y_4$ from the lateral vibrations developed on bearings and six angular rotations as follows: driving motor $\theta_m$, pinion $\theta_1$, wheel $\theta_2$, pinion $\theta_3$, wheel $\theta_4$, and load $\theta_L$ from the torsional vibrations.

By using the principle of virtual work, the equation of motion for the two-stage spur gear system can be derived by taking the derivative of the LaGrange function with respect to the lateral and angular displacement of the gears:

$$m_1\ddot{y}_1 + c_1\dot{y}_1 + c_{t_1}\left[\left(\dot{y}_1 + R_1\dot{\theta}_1\right) - \left(\dot{y}_2 + R_2\dot{\theta}_2\right)\right] + k_1 y_1 + k_{t_1}[(y_1 + R_1\theta_1) - (y_2 + R_2\theta_2)] = 0 \tag{45}$$

$$m_2\ddot{y}_2 + c_2\dot{y}_2 - c_{t_1}\left[\left(\dot{y}_1 + R_1\dot{\theta}_1\right) - \left(\dot{y}_2 + R_2\dot{\theta}_2\right)\right] + k_2 y_2 - k_{t_1}[(y_1 + R_1\theta_1) - (y_2 + R_2\theta_2)] = 0 \tag{46}$$

$$m_3\ddot{y}_3 + c_3\dot{y}_3 - c_{t_2}\left[\left(\dot{y}_3 + R_3\dot{\theta}_3\right) - \left(\dot{y}_4 + R_4\dot{\theta}_4\right)\right] + k_3 y_3 + k_{t_2}[(y_3 + R_3\theta_3) - (y_4 + R_4\theta_4)] = 0 \tag{47}$$

$$m_4\ddot{y}_4 + c_4\dot{y}_4 - c_{t_2}\left[\left(\dot{y}_3 + R_3\dot{\theta}_3\right) - \left(\dot{y}_4 + R_4\dot{\theta}_4\right)\right] + k_4 y_4 - k_{t_2}[(y_3 + R_3\theta_3) - (y_4 + R_4\theta_4)] = 0 \tag{48}$$

$$J_1\ddot{\theta}_1 - c_p\left(\dot{\theta}_m - \dot{\theta}_1\right) + c_{t_1}\left(R_1\dot{y}_1 + R_1^2\dot{\theta}_1 - R_1\dot{y}_2 - R_1 R_2\dot{\theta}_2\right) + k_{t_1}\left(R_1 y_1 + R_1^2\theta_1 - R_1 y_2 - R_1 R_2\theta_2\right) - k_p(\theta_m - \theta_1) = 0 \tag{49}$$

$$J_2\ddot{\theta}_2 + k_g(\theta_2 - \theta_b) + c_{t_1}\left(R_2\dot{y}_1 + R_1R_2\dot{\theta}_1 - R_2\dot{y}_2 - R_2^2\dot{\theta}_2\right) - \\ k_{t_1}\left(R_2 y_1 + R_1 R_2\theta_1 - R_2 y_2 - R_2^2\theta_2\right) + c_g\left(\dot{\theta}_2 - \dot{\theta}_L\right) = 0 \tag{50}$$

$$J_3\ddot{\theta}_3 + c_{t_2}\left(R_3\dot{y}_3 - R_3 R_4\dot{\theta}_4 - R_4\dot{y}_4 + R_3^2\dot{\theta}_3\right) + k_{t_2}\left(R_3^2\theta_3 + R_3 y_3 - R_4 y_3 + R_3 R_4\theta_4\right) = 0 \tag{51}$$

$$J_4\ddot{\theta}_4 + k_g(\theta_4 - \theta_L) - c_{t_2}\left(R_4\dot{y}_3 + R_3 R_4\dot{\theta}_3 - R_4\dot{y}_4 - R_4^2\dot{\theta}_4\right) - \\ k_{t_2}\left(R_4 y_3 + R_3 R_4\theta_3 - R_4 y_4 - R_4^2\theta_4\right) + c_g\left(\dot{\theta}_4 - \dot{\theta}_L\right) = 0 \tag{52}$$

$$J_m\ddot{\theta}_m + k_p(\theta_m - \theta_1) + c_p\left(\dot{\theta}_m - \dot{\theta}_1\right) = T_m \tag{53}$$

$$J_L\ddot{\theta}_L - k_g(\theta_4 - \theta_L) - c_g\left(\dot{\theta}_4 - \dot{\theta}_L\right) = -T_L \tag{54}$$

Finally, the matrix can be expressed as follows when a pit and crack on the root and surface of a pinion tooth coexist:

$$[M]\{\ddot{q}\} + [C]\{\dot{q}\} + [K]\{q\} = \{F\} \tag{55}$$

### 3.2. Numerical Analysis of a Two-Stage Gearbox System with Coupled Pitting and Cracking on the Tooth Surface

The advanced RPM–Frequency diagnostic technique will be used in the simulation to effectively identify features of combined faults in a spur gear system [24]. The equations will be solved using a MATLAB solver with the ode45 subroutine and a Runge-Kutta discretization approach. For this study, slight pitting has been considered negligible at the early stage. The simulation will last for ten seconds for each analysis, and the results will be discussed in terms of the simulated and gear responses:

Figure 7a illustrates the vibration response of a gearbox in good condition with no defects in the lateral displacement. The vibration signal is stable due to the perfect gear transmission.

In Figure 7b, the frequency of the gear mesh (50.06 Hz) is the dominant amplitude in the frequency spectrum (39.08 Hz) in the healthy gearbox. There are no harmonics or small sideband amplitudes present around either frequency.

Figure 8a illustrates a drastic increase in peak amplitudes over a period of 4 to 10 s. The size of the pitting defect on the tooth root is proportionate to the change in measured peak amplitudes.

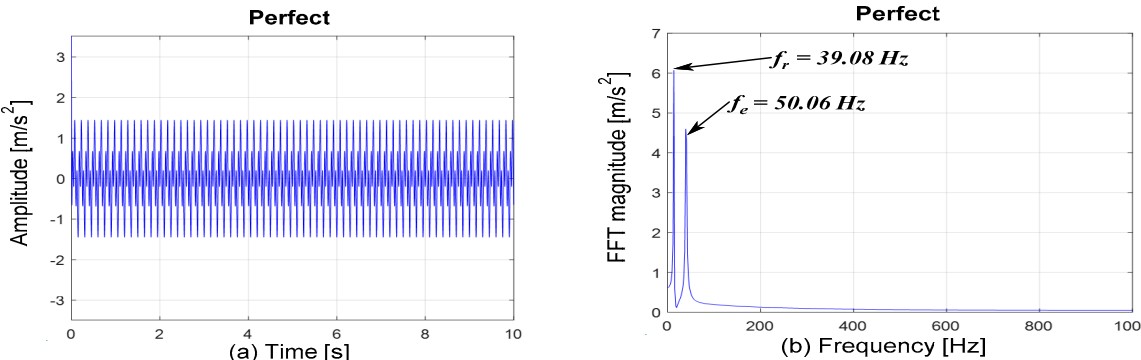

**Figure 7.** Healthy gear response: (**a**) Time-domain and (**b**) FFT.

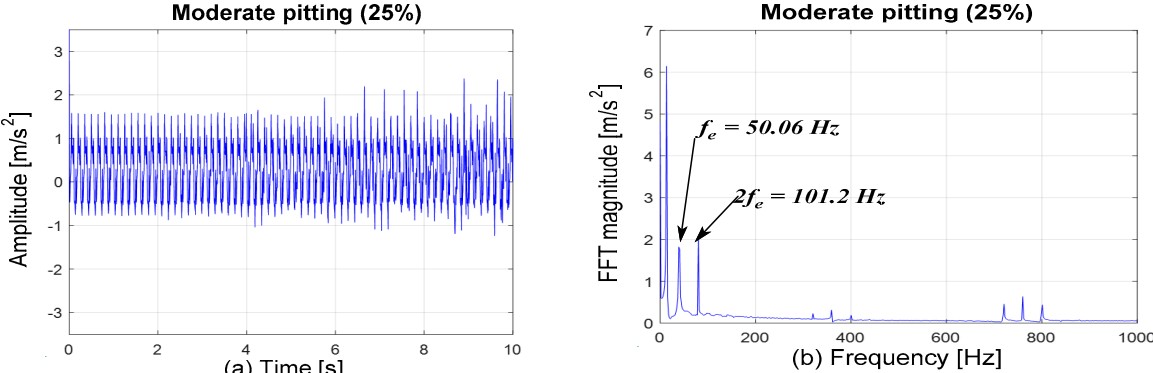

**Figure 8.** Gear vibration response with a pitted tooth: (**a**) Time-domain and (**b**) FFT.

In Figure 8b, there is a noticeable decrease in the amplitude of the gear meshing frequency from 4.6 to 1.839 m/s$^2$. Additionally, three sub-harmonics, a 25% pitting, and sidebands in the frequency range of 80.08 to 800 Hz are present.

Figure 9a displays the highest fluctuation (between 0 and 10 s), with the primary amplitude peak ranging from 1.5 to 2.48 m/s$^2$ between 4 and 8 s. The fluctuation in peak amplitudes suggests the onset of crack-induced tooth surface damage at 25%.

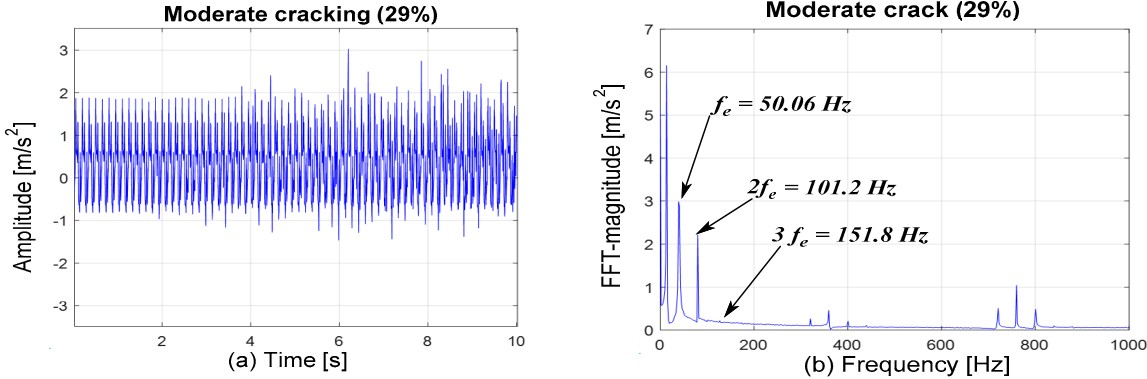

**Figure 9.** Gear vibration response with cracked tooth: (**a**) Time-domain and (**b**) FFT.

In Figure 9b, the propagation of cracks on the tooth root gear causes a fluctuation and decrease in the amplitudes of the harmonics and sidebands, as well as a variation in the meshing frequency of the gears ranging from 4.6 to 3 m/s$^2$.

In Figure 10a, it can be observed that, as the pitting zone increases in size, the depth of the cracks increases significantly, the peak amplitudes of the vibration increase, and the structure becomes unstable within 2 to 10 s.

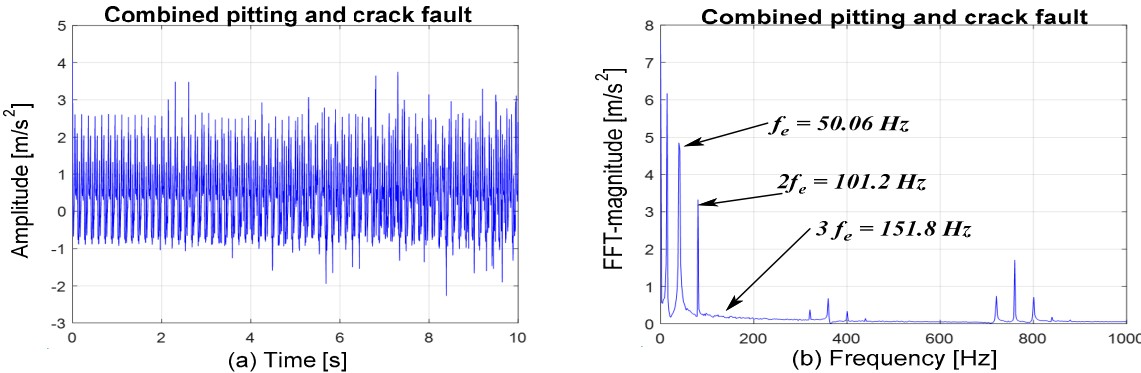

**Figure 10.** Gear with pitted–cracked tooth vibration response: (**a**) Time-domain and (**b**) FFT.

In Figure 10b, due to the pitting cracking defect, the presence of three sub-harmonics surrounded by sidebands in the range of 80.08 to 800 Hz is observed. The gear meshing frequency (*fe*) increases significantly from 4.6 to 4.839 m/s$^2$.

Figure 11a displays a sample spectrum at 600 working speeds, which produces a wide range of vibration frequencies when the gear is in good condition. As gear faults progress, the narrowband components increase in amplitude (fluctuation of peaks) and serve as a useful indicator for detecting suspected gearbox failures such as cracks, as shown in Figure 11b. In Figure 11c, the frequency range of vibration, peak size, shape, and the onset of narrow-band processes change in response to the pitting impact on the surface of the tooth. Meanwhile, as shown in Figure 11d, the coexistence of pitting and cracking causes a significant duplication of energy frequency between 100 Hz and 300 Hz and between 300 Hz and 450 Hz, indicating the evolution of chaotic motion in the spur gear system. As a result, it is observed that the vibration peaks are visible in the spectrum once during each rotation when the faulty gear's surface and the healthy gear's surface engage.

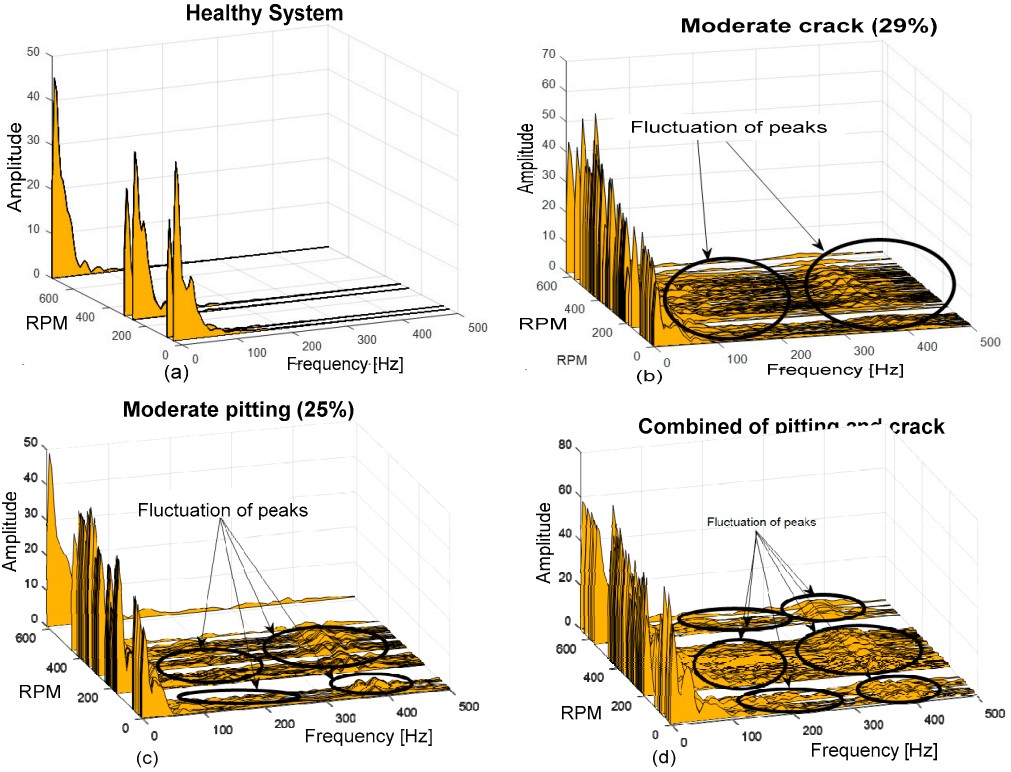

**Figure 11.** 3D-Waterfall: (**a**) healthy gear, (**b**) cracked gear, (**c**) pitted gear, (**d**) pitted/cracked gear.

## 4. Experimental Model

In the dynamic analysis of a two-stage spur gear, extensive studies have been conducted with a crack and a pit running at a constant speed, respectively. To evaluate the simulation model for maximum speed conditions, a two-stage spur gear test rig was built. As shown in Figure 12, the experiments were conducted on a gearbox test bench. The test spur gear was lubricated with grease before being installed on the shafts held by bearings, as specified in the manufacturer's brochure. The experimental set includes an electric motor with a jaw-type coupling to the gearbox and a speed regulator to control the rotation frequency of the gearbox input shaft. The rotational speed, acceleration, and acting load of the system, which has a maximum speed of 1420 rpm, were measured using the test bench.

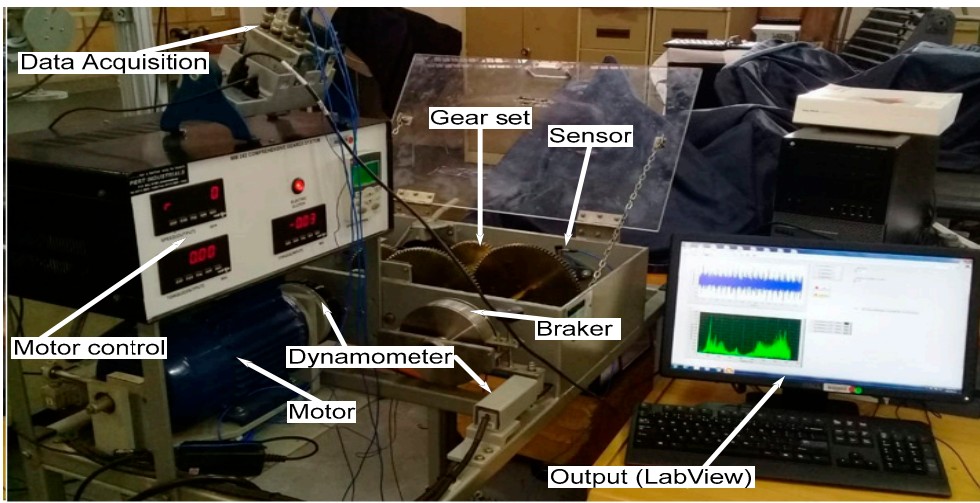

**Figure 12.** Experimental test rig.

A 0.55 kW three-phase induction motor drives a pinion positioned on the input shaft. A resistive torque is provided by the mechanical brake dynamometer, which is coupled to the output shaft of the transfer case, where the reducer is placed, and a power supply with the necessary data-gathering system. Two small piezoelectric accelerometer probe sensors measure vibration signals in the gear system, with lateral and vertical direction (Y) sensitivity of 2.56 mv/ms$^{-2}$ and an acceleration sensor range of 200 g to 200 g.

For an experimental study, a thin saw cut at the tooth root simulates a cracked gear tooth to determine the crack depth, as shown in Figure 13a. A scanning electron microscope is used to measure the cracking length pitting in Figure 13b and the combined fault pitting and crack diameters in Figure 13c.

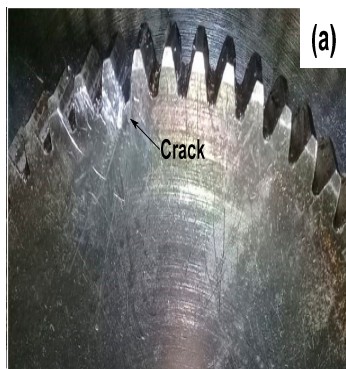 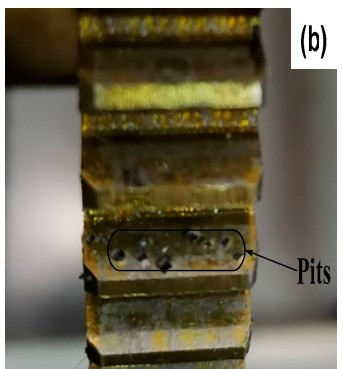 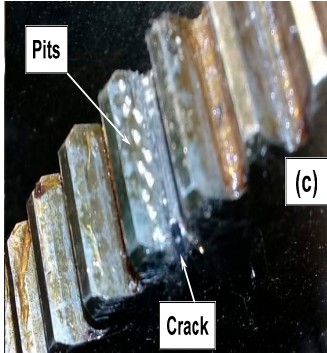

**Figure 13.** (**a**) Gear with crack, (**b**) Gear with pits, (**c**) Pitted–cracked gear.

To create the comprehensive gear system, the Laboratory Virtual Instrument Engineering Workbench (LabVIEW) was used. LabVIEW-based data acquisition involves building software to collect data from various sensors. The National Instruments (NI) 9234 vibration acquisition card was chosen for the analog input modules. For the experiments, the sampling rate is 10 kHz, and each sample lasts eight seconds to measure the signals within a frequency range of 0 to 15 kHz with a frame rate of 1500.

The specifications of a spur gear system are provided in Table 1, and these data are used to compute the meshing stiffness and dynamic reactions of the gear system with coupled pitting and cracking on the tooth surface. The effects of coupled pitting and cracking on the tooth surface on mesh stiffness and vibration characteristics are then studied and discussed.

### 4.1. Experimental Results

The experimental model results on the dynamic properties of pitted and cracked gears are examined to confirm the simulated results using the proposed method. The first test bench is used to evaluate the dynamic behavior of a two-stage spur gear with a moderated crack of 0.5 mm in depth, a moderated pitting gear, and their coexistence in real-world conditions.

Table 600. rpm, a frequency of 10 Hz, and, subsequently, a gear mesh frequency (*fe*) of 50.97 Hz make it possible to visualize the evolution of the harmonic amplitude as a function of frequency. The results of the experiments shown in Figures 14–17 demonstrate the capability of diagnosing gear failures in real machines using large amplitude fluctuations in fault frequencies caused by sidebands. The vibration of the healthy gear at rest contains a noise component due to the impact behavior of the teeth and other mechanical components such as shafts and bearings, as shown in Figure 14a,b.

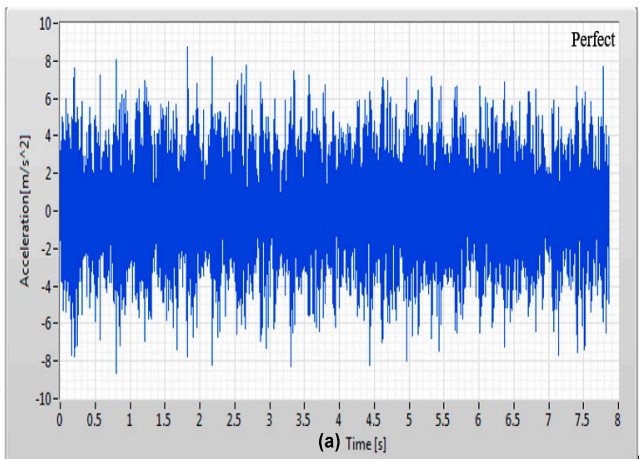 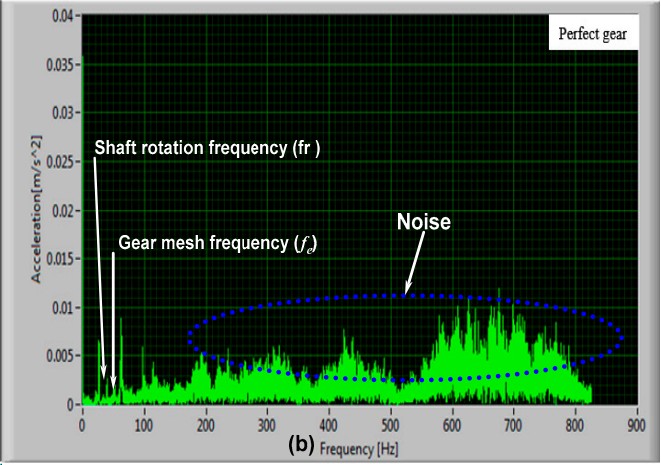

**Figure 14.** Responses of a healthy gear: (**a**) Time-domain (**b**) Spectrum (FFT).

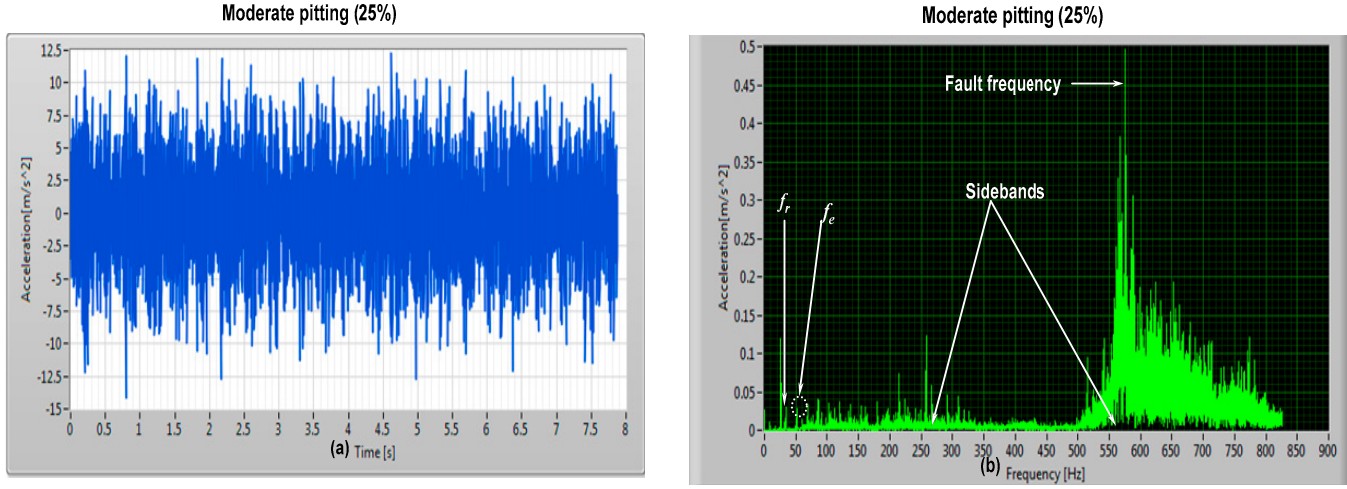

**Figure 15.** Responses of faulty gear: (**a**) Time-domain and (**b**) Spectrum (FFT).

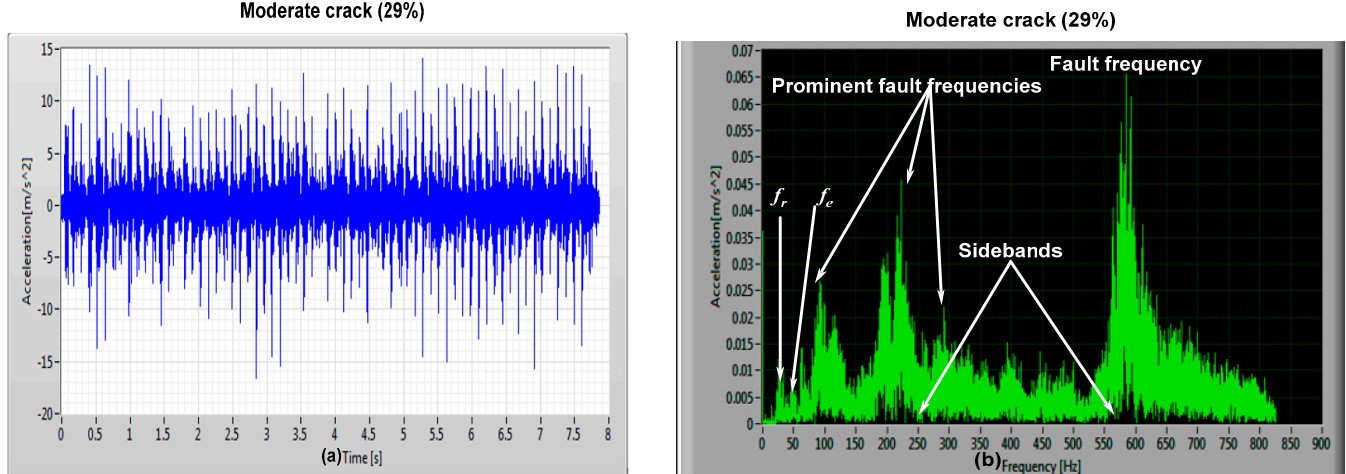

**Figure 16.** Responses of faulty gear: (**a**) Time domain and (**b**) Spectrum (FFT).

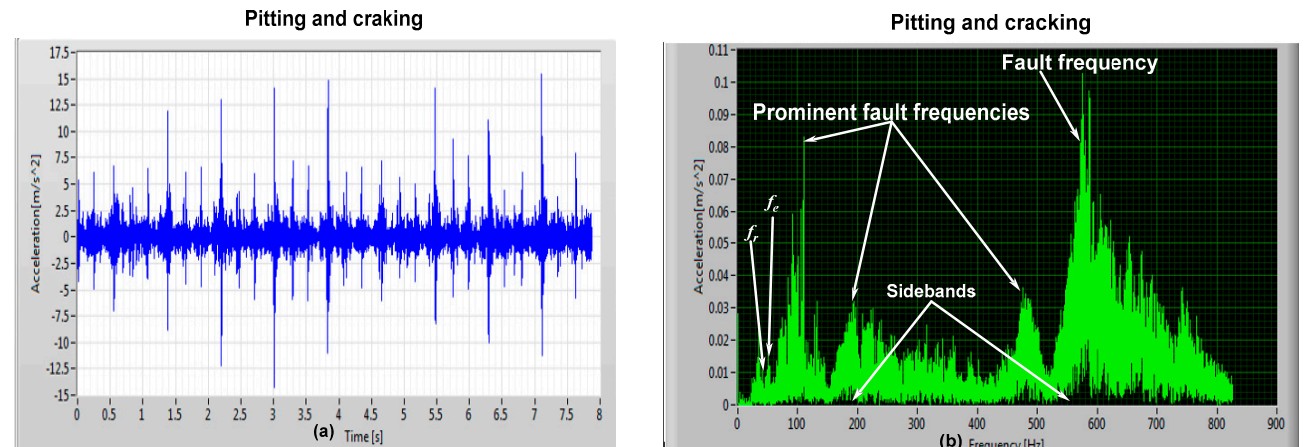

**Figure 17.** Responses of the faulty gear with the pitted–cracked tooth. (**a**) Time domain and (**b**) FFT.

The vibration signals measured during the meshing of the pitted spur gear are shown in Figure 15a. In comparison to a normal state, as shown in Figure 14a, it is observed that the vibration's amplitude is higher.

Figure 15b illustrates the spectra of the measured signals, which show the appearance of a peak near 588 Hz, which indicates the presence of pitting. As seen in Figure 15b, the sidebands surrounding the gear meshing frequency (*fe*) are quite noticeable compared to the normal state, which implies that a gear tooth has been damaged and helps us to understand the root cause of the increased vibration.

Figure 16a depicts the vibration signals measured during the meshing of the cracked spur gear. In comparison to a normal state, as shown in Figure 14a, it is observed that the vibration's amplitude is higher.

Figure 16b depicts the spectra of the measured signals, which show the appearance of a peak near 588 Hz, which indicates the presence of pitting. There is a noticeable increase in the amplitude of the sidebands surrounding the gear meshing frequency (*fe*) and along the frequency range of 100 to 588 Hz compared to the pitting case in Figure 15b.

The result of vibration acceleration measurement for a gear with pitting and cracking coexistence conditions (Figure 17a) demonstrates a considerable increase in vibration amplitude compared to the results of Figures 14a–16a independently. The coexistence of a 0.5-mm crack defect and 13 pits significantly impacts the vibration signal amplitude, as shown in Figure 17b. However, as shown in Figure 18, the frequency plot of the experimental results obtained using the 3-D waterfall frequency shows a sample spectral for 50,000 operating speeds, indicating that a wide range of vibration frequencies is generated.

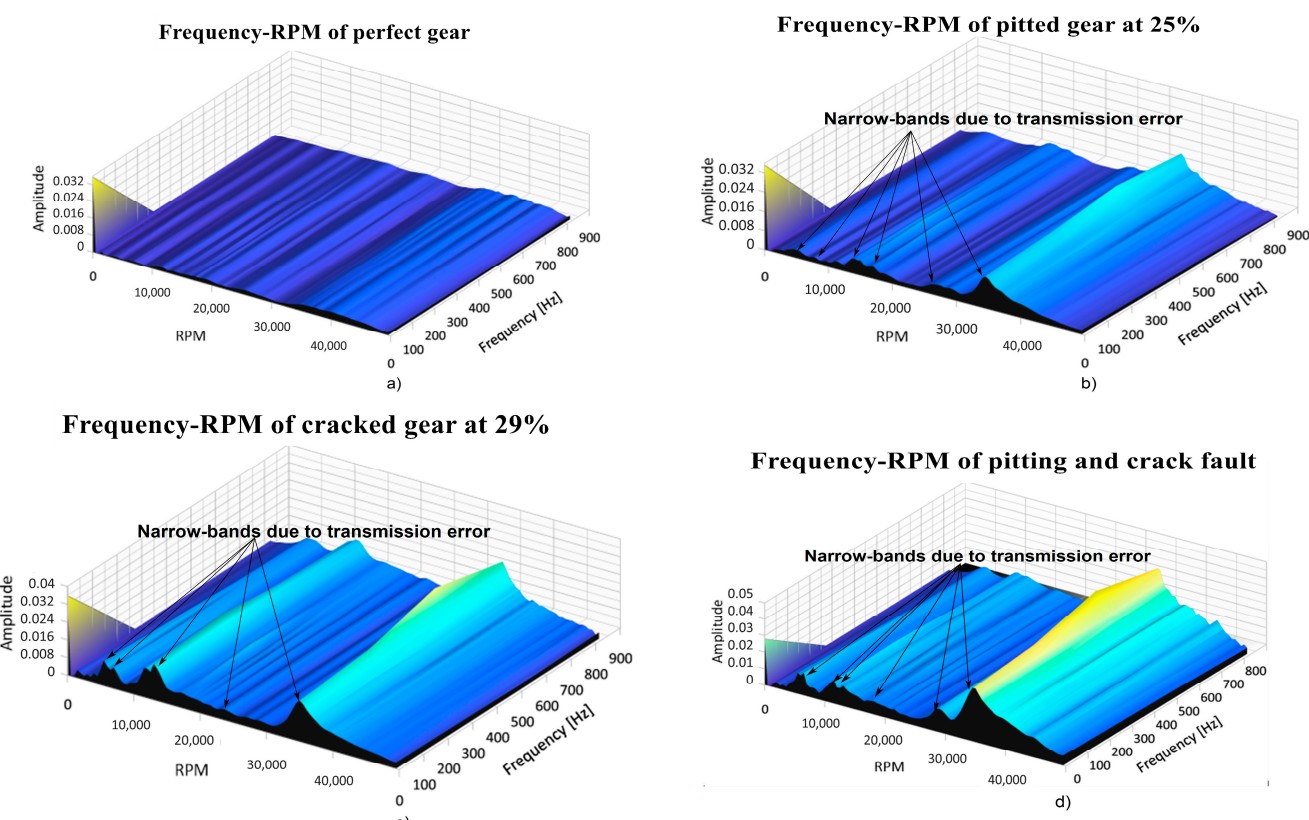

**Figure 18.** Spectrogram of gear: (**a**) perfect, (**b**) cracked tooth, (**d**) pitted tooth, and (**c**) pitted–cracked tooth.

Due to the high number of erroneous peaks and signal leaks, spectral and time analysis of the waveform alone are insufficient for diagnosing the problem. This deficiency is addressed by extracting non-stationary data from the experimental test using the RPM–frequency mapping technique.

### 4.2. Feature Extraction of Experimental Results

In Figure 18a, the spectrogram shows that the energy bands have a small amplitude, which indicates that the gear is not broken. In the presence of a crack endangering the tooth root, Figure 18b shows how the size and shape of the bands alter, and the narrow-band process begins. Three bands with high amplitudes and frequencies ranging from 0 to 900 Hz are depicted in Figure 18c. In Figure 18d, the RPM range is divided into four dominant band segments used to monitor the evolution of vibration signal amplitudes, identify pitting and cracking features, and distinguish the most dominant fault.

In Figure 18d, the RPM range is divided into four dominant band segments that are used to monitor the evolution of vibration signal amplitudes, identify pitting and cracking features, and distinguish the most dominant fault.

To further analyze the impact of coexisting pitting and cracking on the dynamic response of gear systems, the vibration signal of the damaged gear system has been denoised, as shown in Figure 19.

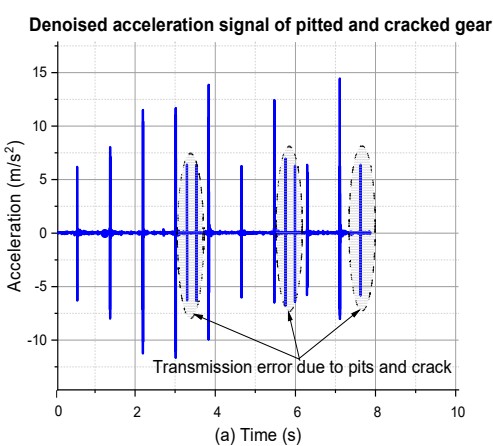 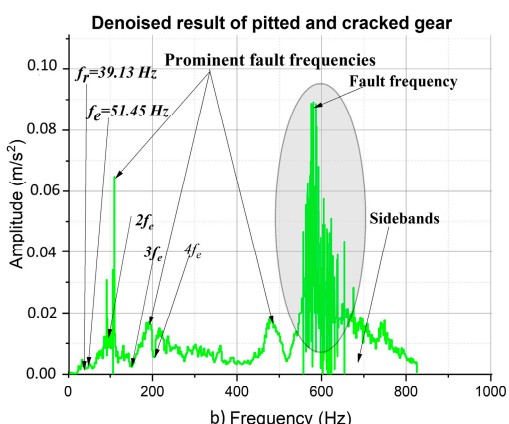

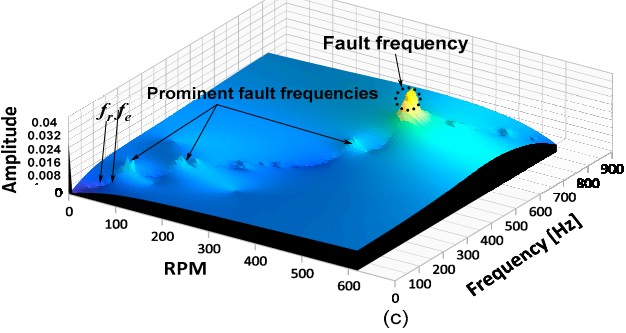

**Figure 19.** Experimental denoised results. (**a**) Time−domain, (**b**) FFT and (**c**) RPM−Frequency.

Figure 19a shows how the gear vibration signals have changed. Because of the simultaneous occurrence of the tooth surface pits and root cracks, the three indicated periodic impulses in the time domain can be identified.

The modulation effect of the fluctuating vibration signal impacted by the coexistence of pitting and cracking can be noticed from the FFT spectrum in Figure 19b, and it is observed that the characteristic frequencies ($f_r$ and $f_e$) are significantly affected. It has been observed that the sidebands and harmonic amplitudes of pitted and cracked gears can be directly identified from the RPM–frequency map, as shown in Figure 19c.

## 5. Discussion

According to the simulated results, as shown in Figure 6, the gear mesh stiffness from the simulated results appears as a periodic decrease with a mesh frequency corresponding to the defective gear rotation.

From the obtained vibration signals of the spur gear system, as shown in Figures 11 and 18, a cracked tooth has a much higher vibration amplitude than a pitted defect. The impact characteristics of the vibration signals, such as the gear mesh frequency and the rotation frequency of the pinion shaft, obtained by the simulation model, as shown in Figures 7–11, are more obvious than those of the experimental model. Due to the complexity of the experimental conditions, it is not unavoidable that external interference may alter the results, such as the noise and other mechanical elements (bearings and shaft) of the gear system, which scrambled the vibration signal shown in Figures 14–18. As a result, there are some experimental variations. However, most of the concepts still hold, showing that the crack characteristics have a greater impact on the vibration signal than the pitting characteristics.

## 6. Conclusions

In summary, this study proposes a novel model of tooth pitting–crack coexistence faults in gear transmission systems and a method of calculating the stiffness of a spur gear with a combined pitting and crack fault. The influence of the crack and pitting parameters and time-varying mesh stiffness on the vibration characteristics of a gear system is studied. The coexistence of pitting and cracking faults on a two-stage spur gear system is analyzed. Both the time domain and frequency domain results and the RPM-frequency map are obtained for the 10 degrees of freedom gear system with coexistence defects and compared with experimental results. Based on the operating condition of a 10 Hz input frequency, several conclusions are obtained:

(1) Cracking predominates in terms of meshing stiffness when the crack is moderate (29%) and the pitting is moderate (25%).

(2) By observing the vibration mutation in the acceleration time domain, the vibration peaks can be used to identify a defect; for the acceleration signal in the frequency domain, the sideband frequency rise can indicate the nature of the defect.

(3) The sideband in the RPM–frequency map responses is more sensitive to cracking and pitting coupled faults affecting the gear tooth surface than the time domain and FFT responses of the dynamics spur gear transmission system. The surface crack fault is easier to diagnose than the surface pit fault because the side frequencies of the dynamics spur gear transmission system increase and change more quickly and drastically during the surface crack propagation.

(4) According to the simulation results for a two-stage spur gear transmission system with a tooth cracking and pitting couple, complex sidebands form close to the gear mesh frequency, and their harmonics and amplitudes rise as the severity of the cracking and pitting couple increases. The experimental results indicate cracking and pitting on the teeth with detailed defect characteristics which qualitatively assess the accuracy of the modeling results.

**Author Contributions:** K.H.Y.H.: conceptualization, methodology, software, validation, formal analysis, data curation—original draft, writing—review, and editing; B.X.T.K.: conceptualization, metho-dology, validation, formal analysis, data curation, review, and editing—original draft and supervision; A.A.A.: review and editing. All authors have read and agreed to the published version of the manuscript.

**Funding:** This research received no external funding.

**Institutional Review Board Statement:** Not applicable.

**Data Availability Statement:** Not applicable.

**Acknowledgments:** The authors are grateful for the resources and equipment provided by the Department of Industrial Engineering, Operation Management, and Mechanical Engineering at Vaal University of Technology (South Africa) to enable this work.

**Conflicts of Interest:** The authors declare no conflict of interest.

**Abbreviations**

The following abbreviations are used in this manuscript:

| | |
|---|---|
| $m_1$ | Pinion mass at the first stage |
| $m_2$ | Wheel (gear) mass at the first stage |
| $m_3$ | Pinion mass at the second stage |
| $m_4$ | Gear mass at the second stage |
| $x_1$ | The linear displacement of the pinion at the first stage |
| $x_2$ | The linear displacement of the wheel at the second stage |
| $x_1$ | The linear displacement of the pinion at the first stage |
| $x_2$ | The linear displacement of the wheel at the second stage |
| $y_1$ | The linear displacement of the pinion at the first stage |
| $y_2$ | The linear displacement of the wheel at the first stage |
| $y_3$ | The linear displacement of the pinion at the second stage |
| $y_4$ | The linear displacement of the wheel at the second stage |
| $J_1$ | Mass moment of inertia of the pinion at the first stage |
| $J_2$ | Mass moment of inertia of the gear at the first stage |
| $J_3$ | Mass moment of inertia of the pinion at the second stage |
| $J_4$ | Mass moment of inertia of the gear at the second stage |
| $J_m$ | Mass moment of inertia of the motor |
| $J_L$ | Mass moment of inertia of the load |
| $k_{y_1}$ | Stiffness of the input bearing in **y**-direction at the first stage |
| $k_{y_2}$ | Stiffness of the input bearing in **y**-direction at the second stage |
| $k_{y_3}$ | Stiffness of the output bearing in **y**-direction at the first stage |
| $k_{y_4}$ | Stiffness of the output bearing in **y**-direction at the second stage |
| $k_p$ | Torsional stiffness of the input shaft coupling |
| $k_g$ | Torsional stiffness of the output shaft coupling |
| $k_t$ | Gear meshing stiffness |
| $c_g$ | Torsional damping of the output shaft coupling |
| $c_p$ | Torsional damping of the input shaft coupling |
| $c_t$ | Gear meshing damping |
| $\theta_1 \ and \ \theta_2$ | The angular displacement of the pinion and gear at the first stage |
| $\theta_3 \ and \ \theta_4$ | The angular displacement of the pinion and gear at the second stage |
| $R_1 \ and \ R_2$ | Base circle radius of pinion and wheel at the first stage |
| $R_3 \ and \ R_4$ | Base circle radius of pinion and wheel at the second stage |
| $C_{y_1} \ and \ C_{y_2}$ | Damping of the input bearing and output bearing at the first stage |
| $C_{y_3} \ and \ C_{y_4}$ | Damping of the input bearing and output bearing at the second stage |

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
