# Peer review of "Influence of Coexistence of Pitting and Cracking Faults on a Two-Stage Spur Gear System"

_vibration, doi:10.3390/vibration6010013_

Round 1

Reviewer 1 Report

This paper covers the influence of coexistence of pitting and cracking faults on a two-stage spur gear system. The following comments are suggested.

 (1)  Whole: A part of notations is out of alignment above.

(2)  p. 6, Figure 3: The explanation of (a) is missing.

(3)  p. 6, line 2 below: The word inside the parentheses is missing.

(4)  p. 7, Figure 4: The explanation is missing.

(5)  p. 8, Equations (17) and (18): Other words except English include.

(6)  Trigonometric function is not italic.

(7)  The number is not italic.

(8)  The unit is not italic.

(9)  p.13, line 7 below: Figure 12 seems to be a mistake.

(10)   p. 14, line 2: The number after “Figure” is missing.

(11)    p. 14, Figure 6: FFT analyzer seems to be used in experiments. This should be shown in figure, and company and model number should be described in text if so.

(12)   p. 15, Figure 7: The explanation of (c) is missing.

(13)   The whole: The magnitudes of pitting and cracking have effects on the vibration of gear pair. This point should be considered.

Author Response

Response to Reviewer 1 Comments

The authors would like to thank the manuscript reviewer for their constructive and professional comments and suggestions, which were very helpful in improving the paper. Additional explanations and analyses were introduced in the amended version of the article.

The whole presentation of the manuscript changed during the revisions of the work and has affected the page numbers of all sections and figure numbers.

Please, find below our responses to all the comments.

Point 1:  Whole: A part of the notations is out of alignment above.

 Response 1: from p.4 to p.10. All the notations have been aligned accordingly.

Point 2: p. 6, Figure 3: The explanation of (a) is missing.

Response 2: p.6 changed to p.5 and Figure 3 changed to Figure 2 after revision of the manuscript. P.6, Figure 2, the study of the coexistence of pitting and cracking in gears starts by showing the path of crack propagation at the root of the tooth, defined as a straight line in Figure 2(a), and then the normal distribution of pitting along the tooth profile direction and the uniform distribution of pitting along the tooth width (L) direction, as shown in Figure 2(b).

Point 3:  p. 6, line 2 below: The word inside the parentheses is missing.

 Response 3: p.5, the whole paragraph has been rewritten as follows: The study of the coexistence of pitting and cracking in gears begins by illustrating the path of crack propagation at the root of the tooth, defined as a straight line in Figure 2(a), and then the normal distribution of pitting along the tooth profile direction as well as the uniform distribution of pitting along the tooth width (L) direction, as shown in Figure 2(b).

Point 4: p. 7, Figure 4: The explanation is missing.

Response 4: p.7 changed to p.6 and Figure 4 change to Figure 3 after revision of the manuscript.

p.5-6, Furthermore, the different top views of the affected gear tooth are presented as shown in Figure 3. At the early stage, Figure 3(a) shows the tooth failure region with a constant crack depth, and Figure 3(b) shows the tooth failure region with a constant pit depth. As a result, the coexistence of pitting and cracking is uniformly distributed in Figure 3(c). This changes the effective section of the gear tooth area and the area moment of inertia, as well as the gear meshing stiffness.

Point 5:  p. 8, Equations (17) and (18): Other words except English include.

 Response 5: p.7, Due to the crack influence on the effective moment of inertia and the cross-section of the surface, the bending and shear stiffness will change. As a result, the effective moment of inertia and cross-section of the surface at a distance x from the root of the tooth is determined using equations (17) and (18).

(1)

(2)

where,is the effective moment of inertia of the cracked tooth.is the cross-section of the surface of the cracked tooth. The crack depth called q is only considered if it is less than half the thickness of the tooth base noted h0 and hx is denoted as the height of the section at the lowest contact point E on the tooth.

Point 6: Trigonometric function is not italic.

Response 6: In the whole text, all the trigonometric functions have been changed from italic to style.

Point 7:  The number is not in italic.

 Response 7: p.7 to p.10, In the whole text, all the trigonometric functions have been changed from italic to style.

Point 8: The unit is not italic.

Response 8: p.7 to p.10, Each equation number in the text has been changed from italic to style.

Point 9:  The number is not in italic.

 Response 9: p.7 to p.10, Each equation number in the text has been changed from italic to style.

Point 10: The unit is not italics.

Response 10: p.7 to p.10, Each unit in the text has been changed from italic to style.

Point 11:  p.13, line 7 below: Figure 12 seems to be a mistake.

 Response 11: Figure 12 is the correct number, not 6. After reviewing the research design, the figure shifted to page 17.

Point 12: )   p. 14, line 2: The number after “Figure” is missing.

Response 12: p.17, line 7, the missed number of the Figure was 6. After reviewing the whole manuscript, Figure 6(c) changed to Figure 12(c).

Point 13: p. 14, Figure 6: FFT analyzer seems to be used in experiments. This should be shown in the figure, and the company and model number should be described in the text if so

 Response 13: After reviewing the whole work, p. 14 became p. 17, and Figure 6 changed to Figure 12. The description of the analyzer is stated as follows on p. 17 and 18: To create the comprehensive gear system (Figure 17), the Laboratory Virtual Instrument Engineering Workbench (LabVIEW) was used. LabVIEW-based data acquisition entails building software to collect data from a variety of sensors. The National Instruments (NI) 9234 vibration acquisition card was chosen to be used for the analog input modules. For the experiments, the sampling rate is 10 kHz, and each sample lasts eight seconds to measure the signals within a frequency range of 0 to 15 kHz with a frame rate of 1500. Two small piezoelectric accelerometer probe sensors measure vibration signals in the gear system, with lateral and vertical direction (Y) sensitivity of 2.56 mv/ms-2 and an acceleration sensor range of 200 g to 200 g.

Point 14:   p. 15, Figure 7: The explanation of (c) is missing.

Response 14: After reviewing the whole work, p. 15 became p. 18 and Figure 7 changed to Figure 13: For the experimental study, a thin saw cut at the tooth root simulates a cracked gear tooth to determine the crack depth, as shown in Figure 13(a). The pitting diameters in Figure 13(b) and the combined fault pitting and crack in Figure 13(c) are measured using a scanning electron microscope.

All the reviewer's remarks are better highlighted in the manuscript's text.

Please see attached the revised manuscript version.

Reviewer 2 Report

This is not the first paper with this idea of using vibration signal for determination of gear damage. New papers are trying to see correlation of crack and pitting, which is not relevant. In practice occur pitting or crack. This is important from the aspect that 17 references are not enough for paper with this idea. Please expand the state of the art. 

Paper contains experiment that is based on synthetic generation of case that is good for further analysis. 

Figure 7 is somehow changed by changing proportions - width or height. Please improve this figure since this figure is more important than drawings in Figs 3,4 and 5. I do not see the point to emphasis that the pit are circular and things like "pitting area". In Figure 7b pits are randomly given on the way that I never saw in practice. Pits occur few around the middle of tooth profile and then they start to expand. Good approach here would be to give one than few and in third case many pits that are connected. For researchers would be perfect to discover pitting in early phase. 

Why is so much needed to be explain position and angles for crack and pits in Figures 3,4 and 5? Is it important for further analyse?

Pits are simply not circular. This is one of many images in internet that can easily be found. 

I understand that authors synthetically made small pits in circular form which is the easiest way. But, here is important to include theoretical part about pitting. Please, after Introduction write a part about pitting and cracks on theoretical level. 

"However, the pitting is caused by friction between the tooth surfaces during the gear meshing process." This is not true. And it is not a small mistake. Please read literature on this topic. 

"The effective contact area is reduced, the gear teeth are bent, and the gear transmission system bearing capacity is reduced because of pitting on the surface of the gear teeth only." Gear teeth are bent - Why is this important? It happens always on micro level. Pitting has nothing with it. 

So, in Introduction please write just about analysis of vibrations or other way how to discover damage and in next separate part theoretical part about pitting and cracking. 

Material and Methods: What material is used in this analysis? I see just Young Module and Poissons ratio. 

Author Response

Response to Reviewer 2 Comments

The authors would like to thank the manuscript reviewer for their constructive and professional comments and suggestions, which were very helpful in improving the paper. Additional explanations and analyses were introduced in the amended version of the article.

The whole presentation of the manuscript changed during the revisions of the work and has affected the page numbers of all sections and figure numbers.

Please, find below our responses to all the comments.

Point 1: This is not the first paper with this idea of using a vibration signal for the determination of gear damage. New papers are trying to see the correlation of crack and pitting, which is not relevant. In practice occur pitting or crack. This is important from the aspect that 17 references are not enough for a paper with this idea. Please expand the state of the art. 

Response 1: More references are added for more expansion of the art in the introduction on p.1-3.

  1. Yang, L.-t.; Shao, Y.-m.; Jiang, W.-w.; Zhang, L.-k.; Wang, L.-m.; Xu, J. Effects of Tooth Surface Crack Propagation on Meshing Stiffness and Vibration Characteristic of Spur Gear System. Appl. Sci. 2021,11, 1968. https://doi.org/10.3390/app11041968.
  2. Hou, J.; Yang, S.; Li, Q.; Liu, Y. Effect of a Novel Tooth Pitting Model on Mesh Stiffness and Vibration Response of Spur Gears. Mathematics 2022, 10, 471. https://doi.org/10.3390/math10030471.
  3. Li, Y.; Yuan, S.; Wu, W.; Liu, K.; Lian, C.; Song, X. Vibration Analysis of Two‐Stage Helical Gear Transmission with Cracked Fault Based on an Improved Mesh Stiffness Model. Machines 2022, 10, 1052. https://doi.org/10.3390/ machines10111052.
  4. Xiong, Y.S.; Huang, K.; Xu, F.W.; Yi, Y.; Sang, M.; Zhai, H. Research on the influence of backlash on mesh stiffness and the nonlinear dynamics of spur gears. Appl. Sci. 2019, 9, 1029.
  5. Jiang, F.; Ding, K.; He, G.L.; Sun, Y.L.; Wang, L.H. Vibration fault features of the planetary gear train with cracks under time-varying flexible transfer functions. Mech. Mach. Theory 2021, 158, 104237.
  6. Meng, Z. Vibration response and fault characteristics analysis of gear based on time-varying mesh stiffness. Mech. Mach. Theory 2020, 148, 103786.
  7. Tiancheng Ouyang, Geng Wang, Liang Cheng, Jinxiang Wang, Rui Yang, Comprehensive diagnosis, and analysis of spur gears with pitting-crack coupling faults, Mechanism and Machine Theory, Volume 176, 2022,104968, ISSN 0094-114X.
  8. Feng, K.; Smith, W.A.; Borghesani, P.; Randall, R.B.; Peng, Z. Use of cyclostationary properties of vibration signals to identify gear wear mechanisms and track wear evolution. Mech. Syst. Signal Process. 2020, 150, 107258.

Point 2: Figure 7 is somehow changed by changing proportions - width or height. Please improve this figure since this figure is more important than drawings in Figs 3,4 and 5. I do not see the point to emphasise that the pit are circular and things like "pitting area". In Figure 7b pits are randomly given on the way that I never saw in practice. Pits occur few around the middle of tooth profile and then they start to expand. Good approach here would be to give one than few and in third case many pits that are connected. For researchers would be perfect to discover pitting in early phase. 

Response 2. The shape of pitting on gear systems can vary depending on the cause and severity of the pits. Pitting can appear as small, shallow, circular depressions on the gear teeth. They can also appear as elongated or irregularly shaped depressions, depending on the severity and duration of the pitting. In some cases, the pits may be deep and have sharp edges. Pitting can also occur in a pattern, such as evenly spaced pits around the circumference of the gear according to references [22].

p.17.and 18. For this study, the sligth pitting has be considered negligeable at the early satge. The pitting starts close to the pitch and spreads to other areas as shown in Figures 13(b) and (c) and appears as circular depressions on the gear tooth surface.

Point 3: Why is so much needed to be explain position and angles for crack and pits in Figures 3,4 and 5? Is it important for further analyse?

 Response 3: p.5, Line 1-3. The position and angle of a crack or pit in a gear system are important in this study because they can affect the overall performance and lifespan of the gears. If a crack or pit is not in the correct position or angle, it can cause uneven wear on the gears, which can lead to increased stress and potential failure. Additionally, the position and angle can also affect the smoothness of the gear movement and the efficiency of the system. By understanding the position and angle of a crack or pit, it is easy to make adjustments to ensure the gears are functioning properly and have a longer lifespan.

Point 4: I understand that authors synthetically made small pits in circular form which is the easiest way. But, here is important to include theoretical part about pitting. Please, after Introduction write a part about pitting and cracks on theoretical level. 

 Response 4: p.2-3. The pitting and cracking of gear teeth can be understood by looking at the mechanics of the gear meshing process. When gears mesh, the teeth come into contact with each other at a specific point, called the contact point. At this point, the gears experience high contact stresses and pressure, as the teeth must conform to each other in order to transmit power.

Pitting can occur when these high contact stresses and pressure cause small surface defects, called pits, to form on the gear teeth. These pits can be caused by a variety of factors, including improper lubrication, high loads, and poor surface finish.

Cracking can occur when the high contact stresses and pressure cause small surface cracks to form on the gear teeth, which can then grow and become larger over time. These cracks can be caused by a variety of factors, including improper heat treatment, high loads, and poor surface finish.

Both pitting and cracking can reduce the strength and durability of the gears, and can ultimately lead to gear failure if left unaddressed.

Point 5: "However, the pitting is caused by friction between the tooth surfaces during the gear meshing process." This is not true. And it is not a small mistake. Please read literature on this topic. 

Response 5: p.2. Line 22-23. Yes, pitting is caused by friction between the tooth surfaces during the gear meshing process; however, the cracking expands with each setting load and is located at the tooth root. These pits can be caused by a variety of factors, including improper lubrication, high loads, and poor surface finish. In this study, the type of gear considered is the spur gear where, friction between the gear teeth during meshing can also cause pitting, as the high contact stresses can cause small surface cracks to form, which can then grow and become pits over time.

Point 6: "The effective contact area is reduced, the gear teeth are bent, and the gear transmission system bearing capacity is reduced because of pitting on the surface of the gear teeth only." Gear teeth are bent - Why is this important? It happens always on a micro level. Pitting has nothing with it. 

 Response 6: p.2, Line 21. It is important to analyze the gear tooth surface because it can be bent due to cracks, which can significantly affect the performance of the gear transmission system. Bent gear teeth can cause a number of issues, such as:

  1. Reduced contact area: When gear teeth are bent, the effective contact area between the teeth is reduced. This can lead to increased wear, pits and tear on the gears, as well as increased vibration and noise.
  2. Increased load on the bearings: Bent gear teeth can cause increased loads on the bearings of the gear transmission system, which can lead to premature bearing failure.
  3. Increased vibration and noise: Bent gear teeth can also cause increased vibration and noise, which can be a sign of a malfunctioning gear system.

All these issues can lead to premature failure of the gear transmission system and can result in downtime and costly repairs. Thus it is important to detect and address bent gear teeth as soon as possible.

Point 7: So, in Introduction please write just about analysis of vibrations or other way how to discover damage and in next separate part theoretical part about pitting and cracking. 

Response 7: Revisions have been done in the Introduction, All parts revised are highlighted in red.

All the reviewer's remarks are better highlighted in the manuscript's text.

Please see attached as well the revised version of the manuscript.

Round 2

Reviewer 1 Report

The manuscript was improved.

Author Response

Response to Reviewer 1 Comments

The authors would like to thank the manuscript reviewer for their constructive and professional comments and suggestions, which were very helpful in improving the paper.

Reviewer 2 Report

Dear Author,

Point 3: Why is so much needed to be explain position and angles for crack and pits in Figures 3,4 and 5? Is it important for further analyse?

 Response 3: p.5, Line 1-3. The position and angle of a crack or pit in a gear system are important in this study because they can affect the overall performance and lifespan of the gears. If a crack or pit is not in the correct position or angle, it can cause uneven wear on the gears, which can lead to increased stress and potential failure. Additionally, the position and angle can also affect the smoothness of the gear movement and the efficiency of the system. By understanding the position and angle of a crack or pit, it is easy to make adjustments to ensure the gears are functioning properly and have a longer lifespan.

RESPONSE: I am still not sure what are we getting by using those angles.

"The position and angle of a crack or pit in a gear system are important in this study because they can affect the overall performance and lifespan of the gears. " - Any occur of pitting affect "overall performance and lifespan of the gears"

"If a crack or pit is not in the correct position or angle, it can cause uneven wear on the gears, which can lead to increased stress and potential failure." - What is correct position or angle of pit? Every pit has an efect that could increase wear and it is increasing stress and possibility of potential failure. There is no "good pit" on some good place that do not have this implications.

"Additionally, the position and angle can also affect the smoothness of the gear movement and the efficiency of the system." - Every pitting is affecting smoothness of gear movement and the efficiency of the system regardless of position and angle. 

"By understanding the position and angle of a crack or pit, it is easy to make adjustments to ensure the gears are functioning properly and have a longer lifespan." - What adjusments? You are making sintetic damage to the tooth flank. In reality pitting is random. You cannot make "good pits" by targeting some position and angle. 

Conclusion response: Anyway, from my point this is not relevant for the science and research but it is not a problem if it is in the paper. 

Point 5: "However, the pitting is caused by friction between the tooth surfaces during the gear meshing process." This is not true. And it is not a small mistake. Please read literature on this topic. 

Response 5: p.2. Line 22-23. Yes, pitting is caused by friction between the tooth surfaces during the gear meshing process; however, the cracking expands with each setting load and is located at the tooth root. These pits can be caused by a variety of factors, including improper lubrication, high loads, and poor surface finish. In this study, the type of gear considered is the spur gear where, friction between the gear teeth during meshing can also cause pitting, as the high contact stresses can cause small surface cracks to form, which can then grow and become pits over time.

In the paper: "However,Pitting, on the pittingother hand, is caused by friction between the tooth surfaces during the gear meshing process." - This is simply not true. And the mistake is not small. 

Errichello, R. (2013). Gear Surface Pitting Failure and Pitting Life Analysis. In: Wang, Q.J., Chung, YW. (eds) Encyclopedia of Tribology. Springer, Boston, MA. https://doi.org/10.1007/978-0-387-92897-5_571 Gear Surface Pitting Failure and Pitting Life Analysis | SpringerLink

"Hertzian fatigue is caused by repeated contact stresses that cause surface or subsurface fatigue cracks and detachment of material fragments from gear tooth surfaces."

How is Hertzian contact stress linked to friction??? If you can prove somehow what you wrote, it is a much bigger contribution to the science than what you wrote in paper. 

Conclusion response: If you do not know what to do with sentence: "However,Pitting, on the pittingother hand, is caused by friction between the tooth surfaces during the gear meshing process.", just delete it. It would be problematic for a scientific paper to have sentence with a very problematic claim. 

Author Response

Dear reviewer,
